# KIMI-DEV: AGENTLESS TRAINING AS SKILL PRIOR FOR SWE-AGENTS

**Zonghan Yang**[#*†]**, Shengjie Wang**[#*]**, Kelin Fu**[¶]**, Wenyang He**[§]**, Weimin Xiong**[¶]**,**
**Yibo Liu, Yibo Miao, Bofei Gao**[¶]**, Yejie Wang**[◇]**, Yingwei Ma, Yanhao Li, Yue Liu**[♠]**,**
**Zhenxing Hu, Kaitai Zhang**[¶]**, Shuyi Wang, Huarong Chen, Flood Sung, Yang Liu**[#]**,**
**Yang Gao**[#♡]**, Zhilin Yang, Tianyu Liu**[¶†]

Moonshot AI, [#]THU, [¶]PKU, [§]UCAS, [◇]BUPT, [♠]NUS, [♡]Shanghai Qi Zhi Institute

 GitHub: https://github.com/MoonshotAI/Kimi-Dev
 HuggingFace: https://huggingface.co/moonshotai/Kimi-Dev-72B

## ABSTRACT

Large Language Models (LLMs) are increasingly applied to software engineering (SWE), with SWE-bench as a key benchmark. Solutions are split into SWE-Agent frameworks with multi-turn interactions and workflow-based Agentless methods with single-turn verifiable steps. We argue these paradigms are not mutually exclusive: reasoning-intensive Agentless training induces skill priors, including localization, code edit, and self-reflection that enable efficient and effective SWE-Agent adaptation. In this work, we first curate the Agentless training recipe and present Kimi-Dev, an open-source SWE LLM achieving 60.4% on SWE-bench Verified, the best among workflow approaches. With additional SFT adaptation on 5k publicly-available trajectories, Kimi-Dev powers SWE-Agents to 48.6% pass@1, on par with that of Claude 3.5 Sonnet (241022 version). These results show that structured skill priors from Agentless training can bridge workflow and agentic frameworks for transferable coding agents.

## 1 INTRODUCTION

Recent days have witnessed the rapid development of Large Language Models (LLMs) automating Software-Engineering (SWE) tasks (Jimenez et al., 2023; Yang et al., 2024a; Xia et al., 2024; Anthropic, 2024; Pan et al., 2024; Wang et al., 2025a; Wei et al., 2025; Yang et al., 2025a; Kimi et al., 2025; OpenAI, 2025c). Among the benchmarks that track the progress of LLM coding agents in SWE scenarios, SWE-bench (Jimenez et al., 2023) stands out as one of the most representative ones: Given an issue that reports a bug in a real-world GitHub repository, a model is required to produce a patch that fixes the bug, the correctness of which is further judged by whether the corresponding unit tests are passed after its application. The difficulty of the task (as of the date the benchmark was proposed), the existence of the outcome reward with the provided auto-eval harness, as well as the real-world economic value it reflects, have made the SWE-bench a focal point of the field.

Two lines of solutions have emerged for the SWE-bench task. Agent-based solutions like SWE-Agent (Yang et al., 2024a) and OpenHands (Wang et al., 2025a) take an interactionist approach: Instructed with the necessary task description, a predefined set of available tools, as well as the specific problem statement, the agent is required to interact with an executable environment for *multiple turns*, make change to the source codes, and determine when to stop autonomously. In contrast, workflow-based solutions like Agentless (Xia et al., 2024) pre-define the solving progress as a pipeline, which consists of steps like localization, bug repair, and test composition. Such task decomposition transforms the agentic task into generating correct responses for a chain of *single-turn* problems with verifiable rewards (Guo et al., 2025; Wei et al., 2025; He et al., 2025).

The two paradigms have been widely viewed as mutually exclusive. On the one hand, SWE-Agents are born with higher potential and better adaptability, thanks to the higher degree of freedom of the

---

*Indicates equal contribution. † Joint leads.

multi-turn interaction without the fixed routines. However, it has also proved more difficult to train with such frameworks due to their end-to-end nature (Luo et al., 2025; Cao et al., 2025). On the other hand, Agentless methods offer better modularity and the ease to train with Reinforcement Learning with Verifiable Rewards (RLVR) techniques, but more limited exploration space and flexibility, and difficulty in behavior monitoring as the erroneous patterns appear only in the single-turn long reasoning contents (Pan et al., 2024). However, we challenge the dichotomy from the perspective of training recipe: We argue that Agentless training should not be viewed as the ultimate deliverable, but rather as a way to induce skill priors – atomic capabilities such as the localization of buggy implementations and the update of erroneous code snippets, as well as self-reflection and verification, all of which help scaffold the efficient adaptation of more capable and generalizable SWE-agents.

Guided by this perspective, we introduce Kimi-Dev, an open-source code LLM for SWE tasks. Specifically, we first develop an Agentless training recipe, which includes mid-training, cold-start, reinforcement learning, and test-time self-play. This results in 60.4% accuracy on SWE-bench Verified, the SoTA performance among the workflow-based solutions. Building on this, we show that Agentless training induces skill priors: a minimal SFT cold-start from Kimi-Dev with 5k publicly-available trajectories enables efficient SWE-agent adaptation and reaches 48.6% pass@1 score, similar to that of Claude 3.5 Sonnet (the 20241022 version, Anthropic (2024)). We demonstrate that these induced skills transfer from the non-agentic workflows to the agentic frameworks, and the self-reflection in long Chain-of-Thoughts baked through Agentless training further enable the agentic model to leverage more turns and succeed with a longer horizon. Finally, we also show that the skills from Agentless training generalize beyond SWE-bench Verified to broader benchmarks like SWE-bench-live (Zhang et al., 2025) and SWE-bench Multilingual (Yang et al., 2025c). Together, these results reframe the relationship between Agentless and agentic frameworks: not mutually exclusive, but as complementary stages in building transferable coding LLMs. This shift offers a principled view that training with structural skill priors could scaffold autonomous agentic interaction.

The remainder of this paper is organized as follows. Section 2 reviews the background of the framework dichotomy and outlines the challenges of training SWE-Agents. Section 3 presents our Agentless training recipe and the experimental results. Section 4 demonstrates how these Agentless-induced skill priors enable efficient SWE-Agent adaptation, and evaluates the skill transfer and generalization beyond SWE-bench Verified.

## 2 BACKGROUND

In this section, we first review the two dominant frameworks for SWE tasks and their dichotomy in Section 2.1. We then summarize the progress and challenges of training SWE-Agents in Section 2.2. The background introduction sets the stage for reinterpreting Agentless training as skill priors for SWE-Agents, a central theme developed throughout the later sections.

### 2.1 FRAMEWORK DICHOTOMY

Two paradigms currently dominate the solutions for automating software engineering tasks. Agentless approaches decompose SWE tasks into modular workflows (Xia et al., 2024; Wei et al., 2025; Ma et al., 2025a;b; Xie et al., 2025). Typical workflows consist of bug localization, bug repair, and test generation. This design provides modularity and stability: each step could be optimized separately as a single-turn problem with verifiable rewards (Wei et al., 2025; He et al., 2025). However, such rigidity comes at the cost of flexibility. When encountering scenarios requiring multiple rounds of incremental updates, the Agentless approaches struggle to adapt.

By contrast, SWE-agents adopt an end-to-end, multi-turn reasoning paradigm (Yang et al., 2024a; Wang et al., 2025a). Rather than following a fixed workflow, they iteratively plan, act, and reflect, resembling how human developers debug complex issues. This design enables greater adaptability, but introduces significant difficulties: trajectories often extend over tens or even hundreds of steps, context windows of the LLMs must span over the entire interaction history, and the model must handle exploration, reasoning, and tool use simultaneously.

The dichotomy between fixed workflows (*e.g.*, Agentless) and agentic frameworks (*e.g.*, SWE-Agent) has shaped much of the community's perspective. The two paradigms are often regarded as mutually exclusive: one trades off flexibility and performance ceiling for modularity and stabil-

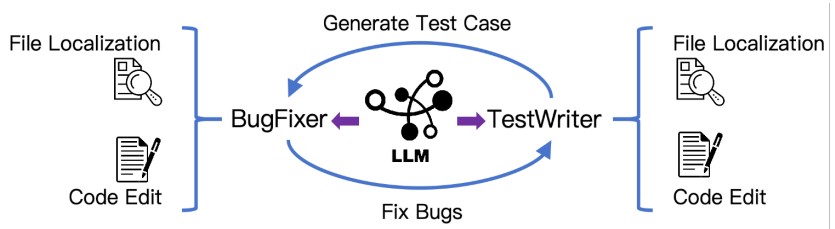

Figure 1: Agentless framework for Kimi-Dev: the duo of BugFixer and TestWriter.

ity, whereas the other makes the reverse compromise. Our work challenges this dichotomy, as we demonstrate that Agentless training induces skill priors that make further SWE-agent training both more stable and more efficient.

## 2.2 TRAINING SWE-AGENTS

Training SWE-agents relies on acquiring high-quality trajectories through interactions with executable environments. Constructing such large-scale environments and collecting reliable trajectories, however, requires substantial human labor as well as costly calls to frontier models, making data collection slow and resource-demanding (Pan et al., 2024; Badertdinov et al., 2024b). Recent studies also attempt to scale environment construction by synthesizing bugs for the reverse construction of executable runtime (Jain et al., 2025; Yang et al., 2025c).

However, credit assignment across long horizons still remains challenging, as outcome rewards are sparse and often only available when a final patch passes its tests. Reinforcement learning techniques have been proposed, but frequently suffer from instability or collapse when trajectories exceed dozens of steps (Luo et al., 2025; Cao et al., 2025). SWE-agent training is also highly sensitive to initialization: starting from a generic pre-trained model often leads to brittle behaviors, such as failing to use tools effectively or getting stuck in infinite loops of specific action patterns (Pan et al., 2024; Yang et al., 2025c).

These limitations motivate our central hypothesis: instead of training SWE-agents entirely from scratch, one can first induce skill priors through agentless training, enhancing the atomic capabilities like localization, repair, test composition, and self-reflection. These priors lay a foundation that makes subsequent agentic training both more efficient and more generalizable.

## 3 AGENTLESS TRAINING RECIPE

Instead of training SWE-agents from scratch, we leverage Agentless training to induce skill priors. Skill priors enhanced by Agentless training include but are not limited to bug localization, patch generation, self-reflection and verification, which lay the foundation for end-to-end agentic interaction. In this section, we elaborate our Agentless training recipe: the duo framework design of BugFixer and TestWriter, mid-training and cold-start, reinforcement learning, and test-time self-play. Sections 3.1–3.4 detail these ingredients, and Section 3.5 presents the experimental results for each of them. This training recipe results in Kimi-Dev, an open-source 72B model that achieves 60.4% on SWE-bench Verified, the SoTA performance among the workflow-based solutions.

## 3.1 FRAMEWORK: THE DUO OF BUGFIXER AND TESTWRITER

In GitHub issue resolution, we conceptualize the process as the collaboration between two important roles: the BugFixer, who produces patches that correctly address software bugs, and the TestWriter, who creates reproducible unit tests that capture the reported bug. A resolution is considered successful when the BugFixer's patch passes the tests provided for the issue, while a high-quality test from the TestWriter should fail on the pre-fix version of the code and pass once the fix is applied.

Each role relies on two core skills: (i) file localization, the ability to identify the specific files relevant to the bug or test, and (ii) code edit, the ability to implement the necessary modifications. For Bug-Fixer, effective code edits repair the defective program logic, whereas for TestWriter, they update precise unit test functions that reproduce the issue into the test files. As illustrated in Figure 1, these two skills constitute the fundamental abilities underlying GitHub issue resolution. Thus, we enhance these skills through the following training recipes, including mid-training, cold-start, and RL.

## 3.2 MID-TRAINING & COLD START

To enhance the model's prior as both a BugFixer and a TestWriter, we perform mid-training with ∼150B tokens in high-quality and real-world data. With the Qwen 2.5-72B-Base (Qwen et al., 2024) model as a starting point, we collect millions of GitHub issues and PR commits to form its mid-training dataset, which consists of (i) ∼50B tokens in the form of Agentless derived from the natural diff patch, (ii) ∼20B tokens of curated PR commit packs, and (iii) ∼20B tokens of synthetic data with reasoning and agentic interaction patterns (upsampled by a factor of 4 during training). The data recipe is carefully constructed to enable the model to learn how human developers reason with GitHub issues, implement code fixes, and develop unit tests. We also performed strict data decontamination to exclude any repository from the SWE-bench Verified test set. Mid-training sufficiently enhances the knowledge in the model about practical bug fixes and unit tests, making it a better starting point for later stages. The details of the recipe are covered in Appendix A.

To activate the model's long Chain-of-Thought (CoT) capability, we also construct a cold-start dataset with reasoning trajectories based on the SWE-Gym (Pan et al., 2024) and SWE-bench-extra (Badertdinov et al., 2024a) datasets, generated by the DeepSeek R1 model (Luo et al. (2025), the 20250120 version). In this setup, R1 acts the roles of Bugfixer and Testwriter, producing outputs such as file localization and code edits. Through supervised finetuning as a cold start with this dataset, we enable the model to acquire essential reasoning skills, including problem analysis, method sketching, self-refinement, and exploration of alternative solutions.

## 3.3 REINFORCEMENT LEARNING

After mid-training and cold-start, the model demonstrates strong performance in localization. Therefore, reinforcement learning (RL) focuses solely on the code edit stage. We construct a training set specifically for this stage, where each prompt is equipped with an executable environment. We further employ multiple localization rollouts from the initial model to generate varied file location predictions, which diversifies the prompts used in code-edit RL.

For the RL algorithm, we adopt the policy optimization method proposed by Kimi k1.5 (Team et al., 2025), which has shown promising results on reasoning tasks in both math and coding. Kimi k1.5 (Team et al., 2025) adopts a simpler policy gradient approach based on the REINFORCE algorithm (Williams, 1992). Similarly to GRPO (Shao et al., 2024), we use the average rewards of multiple rollouts as the baseline to normalize the returns. When adapting the algorithm in our SWE-bench setting, we highlight the following 3 key desiderata:

1. **Outcome-based reward only**: We rely solely on the final execution outcome from the environment as the raw reward (0 or 1), without incorporating any format- or process-based signals. For BugFixer, a positive reward is given if the generated patch passes all ground-truth unittests. For TestWriter, a positive reward is assigned when (i) the predicted test raises a failure in the repo without the ground-truth bugfix patch applied, **AND** (ii) the failure is resolved once the ground-truth bugfix patch is applied.

2. **Adaptive prompt selection**: Prompts with pass@16 = 0 are initially discarded as they do not contribute to the batch loss. This results in an initial prompt set of 1,200 problems and enlarges the effective batch size. A curriculum learning scheme is then applied: once the success rate on the current set exceeds a threshold, 500 new (previously excluded) prompts (with initial pass@16 = 0 but improved under RL) are reintroduced every 100 RL steps to gradually raise task difficulty.

3. **Positive example reinforcement**: As performance improvements begin to plateau in later stages of training, we incorporate the positive samples from the recent RL iterations into the training batch of the current iteration. This approach reinforces the model's reliance on successful patterns, thereby accelerating convergence in the final phase.

**Robust sandbox infrastructure.** We construct the docker environment with Kubernetes (Burns et al., 2016), which provides a secure and scalable sandbox infrastructure and efficient training and rollouts. The infra supports over 10,000 concurrent instances with robust performance, making it ideal for competitive programming and software engineering tasks (see Appendix D for details).

Table 1: Performance comparison for models on SWE-bench Verified under Agentless-like frameworks. All the performances are obtained under the standard 40 patch, 40 test setting (Xia et al., 2024), except that Llama3-SWE-RL uses 500 patches and 30 tests.

| Model | #Params | Resolve Rate (%) |
|---|---|---|
| Llama3-SWE-RL (Wei et al., 2025) | 70B | 41.0 |
| Seed1.5-Thinking (Seed et al., 2025) | 200B | 47.0 |
| OpenAI-o1 (OpenAI, 2024) | - | 48.9 |
| DeepSeek-R1-0120 (Guo et al., 2025) | 671B | 49.2 |
| OpenAI-o3-mini-high (OpenAI, 2025a) | - | 49.3 |
| Claude 3.5 Sonnet (241022) (Anthropic, 2024) | - | 50.8 |
| MiniMax-M1 (Chen et al., 2025a) | 456B | 56.0 |
| DeepSeek-R1-0528 (Guo et al., 2025) | 671B | 57.6 |
| SWE-SWISS (He et al., 2025) | 32B | 58.2 |
| Kimi-Dev (Ours) | 72B | **60.4** |

## 3.4 TEST-TIME SELF-PLAY

After RL, the model masters the roles of both a BugFixer and a TestWriter. During test time, it adopts a self-play mechanism to coordinate its bug-fixing and test-writing abilities.

Following Agentless (Xia et al., 2024), we leverage the model to generate 40 candidate patches and 40 tests for each instance. Each patch generation involves independent runs of the localization and code edit from BugFixer, where the first run uses greedy decoding (temperature 0), and the remaining 39 use temperature 1 to ensure diversity. Similarly, 40 tests are generated independently from TestWriter. For the test patch candidates, to guarantee their validity, we first filter out those failing to raise a failure in the original repo without applying any BugFixer patch.

Denote the rest TestWriter patches as set $\mathcal{T}$, and the BugFixer patches as set $\mathcal{B}$. For each $b_i \in \mathcal{B}$ and $t_j \in \mathcal{T}$, we execute the test suite over the test file modified by $t_j$ for twice: first without $b_i$, and then with $b_i$ applied. From the execution log for the first run, we get the count of the failed and the passed tests from $t_j$, denoted as $\mathrm{F}(j)$ and $\mathrm{P}(j)$. Comparing the execution logs for the two test suite runs, we get the count of the fail-to-pass and the pass-to-pass tests, denoted as $\mathrm{FP}(i,j)$ and $\mathrm{PP}(i,j)$, respectively. We then calculate the score for each $b_i$ with

$$S_i = \frac{\sum_j \mathrm{FP}(i,j)}{\sum_j \mathrm{F}(j)} + \frac{\sum_j \mathrm{PP}(i,j)}{\sum_j \mathrm{P}(j)}, \tag{1}$$

where the first part reflects the performance of $b_i$ under reproduction tests, and the second part could be viewed as the characterization of $b_i$ under regression tests (Xia et al., 2024). We select the BugFixer patch $b_i$ with the highest $S_i$ score as the ultimate answer.

## 3.5 EXPERIMENTS

### 3.5.1 MAIN RESULTS

Table 1 shows the performance of Kimi-Dev on SWE-bench Verified (Jimenez et al., 2023). Instead of the text-similarity rewards used in SWE-RL (Wei et al., 2025), we adopt execution-based signals for more reliable fix quality. Our two-stage TestWriter also improves over prior Agentless systems (Xia et al., 2024; Guo et al., 2025; He et al., 2025), which rely on a single root-level test, by better capturing repository context and mirroring real developer workflows (OpenAI, 2025). Kimi-Dev attains state-of-the-art performance among open-source models, resolving 60.4% of issues.

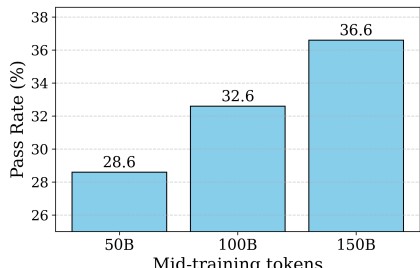

Figure 2: The performance on SWE-bench Verified after mid-training with different training token budgets.

### 3.5.2 MID-TRAINING

In this section, we evaluate the relationship between the amount of data used during mid-training and model performance. Specifically, we finetuned Qwen 2.5-72B-Base with the subset of mid-training data of 50B, 100B, and approximately 150B tokens, and then lightly activated these mid-trained models using the same set of 2,000 Bugfixer

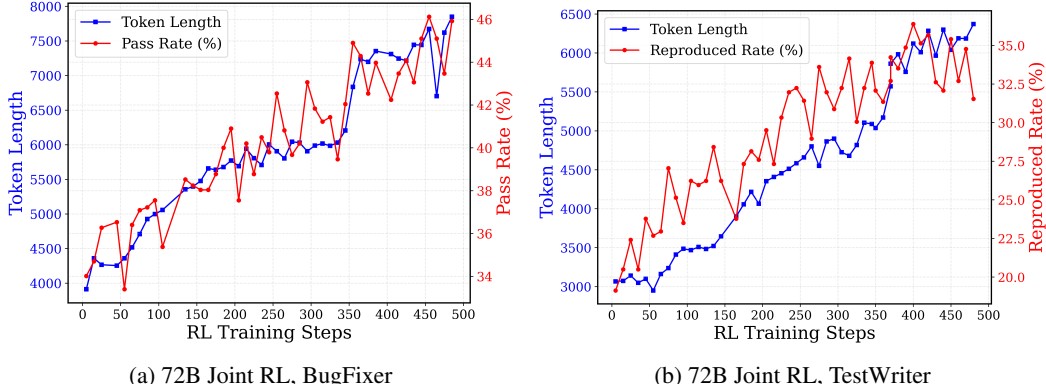

(a) 72B Joint RL, BugFixer

(b) 72B Joint RL, TestWriter

Figure 3: Joint code-edit RL experiments on the model after mid-training and cold-start. The pass rate for BugFixer and the reproduced rate for TestWriter are reported as pass@1 with temperature=1.0. The performance improves consistently as the output becomes increasingly longer.

input-output pairs for SFT cold start. We only report BugFixer pass@1 here for simplicity of evaluation. Figure 2 shows that increasing the number of tokens in mid-training consistently improves model performance, highlighting the effectiveness of this stage.

### 3.5.3 REINFORCEMENT LEARNING

**Experimental setup** We set the training step per RL iteration as 5 and sample 10 rollouts for each of the 1,024 problems from the union of SWE-gym (Pan et al., 2024) and SWE-bench-extra (Badertdinov et al., 2024b). We dynamically adjust the prompt set every 20 iterations to gradually increase task difficulty. We fix the maximum training context length as 64k tokens, since the prompt input contains the contents of the entire files localized by the initial model in advance.

**Results** Figure 3 shows the performance and response length curves on the test set during RL training. The pass rate and the reproduced rate are calculated from pass@1 and temperature=1. Specifically, we observe that both model performance and response length steadily increase, reflecting the expected benefits of RL scaling. Similar RL scaling curves are also observed in our ablation experiments run on Qwen2.5-14B-Instruct models, proving the effectiveness of the RL training recipe across models of different sizes. The experimental details, as well as the ablation studies on positive example reinforcement in Section 3.3, are listed in Appendix C.2). The lengthy outputs consist of in-depth problem analysis and self-reflection patterns, similar to those in the math and code reasoning tasks (Team et al., 2025; Guo et al., 2025). We have also observed that for TestWriter, occasional false-positive examples take place during RL training due to the lack of reproduction coverage. We leave the case studies in Appendix E and further improvement for future work.

### 3.5.4 TEST-TIME SELF-PLAY

Following Section 3.4, we evaluate how the final performance on SWE-bench Verified scales with the number of patches and tests generated. The temperature is fixed at 0 for the initial rollout, and set to 1.0 for the subsequent 39 rollouts. As shown on the left of Figure 4, the final performance improves from 48.0% to 60.4% as the number of patch-test pairs increases from 1×1 to 40×40, and consistently surpasses the results obtained from the majority vote of the BugFixer patches only.

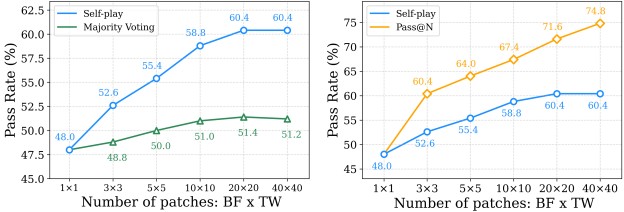

Figure 4: Test-time self-play on SWE-bench Verified. Performance improves with more generated patches and tests. Left: Execution-based self-play consistently surpasses BugFixer majority voting. Right: Self-play performances remain below pass@N where the ground-truth test patch is used, suggesting the room exists for TestWriter to improve.

Table 2: Single-attempt performance of different models on SWE-bench Verified under end-to-end agentic frameworks, categorized by proprietary or open-weight models, and size over or under 100B (as of 2025.09). "Internal" denotes results achieved with their in-house agentic frameworks.

| Model | System | #Params | Pass Rate (%) |
|---|---|---|---|
| *Proprietary* | | | |
| Gemini 2.5 Pro (Comanici et al., 2025) | Internal | - | 60.3 |
| OpenAI-o3 (OpenAI, 2025) | Internal | - | 69.1 |
| GPT-5 (OpenAI, 2025c) | Internal | - | 74.9 |
| Claude 3.5 Sonnet (241022) (Anthropic, 2024) | SWE-Agent | - | 49.0 |
| Claude 3.7 Sonnet (Anthropic, 2025a) | SWE-Agent | - | 62.3 |
| Claude 4.0 Sonnet (Anthropic, 2025b) | SWE-Agent | - | 72.7 |
| *Open Weight, $\geq$ 100B* | | | |
| gpt-oss-120b (High) (OpenAI, 2025b) | Internal | 120B | 62.4 |
| DeepSeek-v3.1 (Guo et al., 2025) | Internal | 671B | 66.0 |
| Kimi-K2-0905 (Kimi et al., 2025) | SWE-Agent | 1T | 69.2 |
| Qwen3-Coder (Yang et al., 2025a) | OpenHands | 480B | 69.6 |
| *Open Weight, < 100B* | | | |
| Openhands-LM (Wang et al., 2025b) | OpenHands | 32B | 37.2 |
| Skywork-SWE (Zeng et al., 2025) | OpenHands | 32B | 38.0 |
| SWE-agent-LM (Yang et al., 2025b) | SWE-Agent | 32B | 40.2 |
| DeepSWE (Luo et al., 2025) | OpenHands | 32B | 42.2 |
| Devstral-Small-2507 (AI & AI, 2025) | OpenHands | 24B | 53.6 |
| gpt-oss-20b (High) (OpenAI, 2025b) | Internal | 20B | 60.7 |
| Kimi-Dev (SFTed) | SWE-Agent | 72B | 48.6 |

Specifically, the self-play result obtained from 3 patches and 3 tests for each instance has already surpassed the performance with majority voting from 40 BugFixer patches. This demonstrates the effectiveness of additional information from test-time execution. The room for improvement of Test-Writer, though, still exists for more powerful self-play: Shown on Figure 4, self-play performances remain below pass@N, where ground-truth test cases serve as the criterion for issue resolution. This finding aligns with Anthropic (2024), which introduced a final edge-case checking phase to generate a more diverse set of test cases, thereby strengthening the role of the "TestWriter" in their SWE-Agent framework. We also report preliminary observations of a potential parallel scaling phenomenon, which requires no additional training and may enable scalable performance improvements. The details of the phenomenon and analyses are covered in Appendix F.

## 4 INITIALIZING SWE-AGENTS FROM AGENTLESS TRAINING

End-to-end multi-turn frameworks, such as SWE-Agent (Yang et al., 2024a; Anthropic, 2024) and OpenHands (Wang et al., 2025a), enable agents to leverage tools and interact with environments. Specifically, the system prompt employed in the SWE-Agent framework (Anthropic, 2024) outlines a five-stage workflow: (i) repo exploration, (ii) error reproduction via a test script, (iii) code edit for bug repair, (iv) test re-execution for validation, and (v) edge-case generation and checks. Unlike Agentless, the SWE-Agent framework doesn't enforce a strict stage-wise workflow; the agent can reflect, transition, and redo freely until it deems the task complete and submits.

The performance potential is therefore higher without a fixed routine; However, the training for SWE-Agent is more challenging because of the sparsity of the outcome reward for long-horizon credit assignment. Meanwhile, our Kimi-Dev model has undergone Agentless training, with its skills of localization and code edit for BugFixer and TestWriter strengthened elaborately. In this section, we investigate whether it can serve as an effective prior for multi-turn SWE-Agent scenarios.

### 4.1 PERFORMANCE AFTER SWE-AGENT FINE-TUNING

We use the publicly available SWE-Agent trajectories to finetune Kimi-Dev. The finetuning dataset we used is released by SWE-smith (Yang et al., 2025b), consisting of 5,016 SWE-Agent trajectories

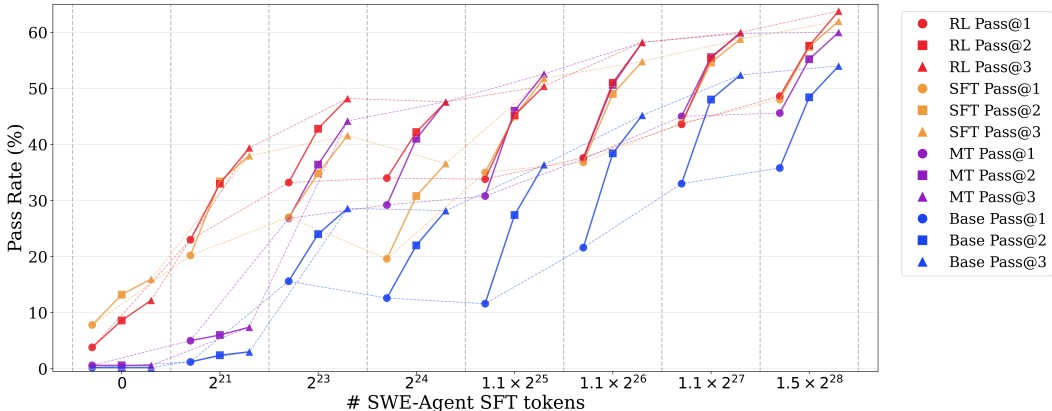

Figure 5: Comparing the quality of the raw Base, the Agentless mid-trained (MT), the Agentless mid-trained with reasoning-intensive cold-start (SFT), and the Kimi-Dev model after RL as the prior for SWE-Agent adaptation. The tokens of the SWE-Agent SFT trajectories are swept over different scales, and the SWE-Agent performances are reported up to pass@3 on SWE-bench Verified.

collected with Claude 3.7 Sonnet (Anthropic, 2025a) in the synthetic environments. We perform supervised fine-tuning over Kimi-Dev, setting the maximum context length as 64K tokens during training, and allowing up to 128K tokens and 100 turns during inference.

As shown in Table 2, without collecting more trajectory data over realistic environments, or conducting additional multi-turn agentic RL, our finetuned model achieves a pass@1 score of 48.6% on SWE-bench Verified under the agentic framework setup, without additional test-time scaling. Using the same SFT data, our finetuned Kimi-Dev model outperforms the SWE-agent-LM (Yang et al., 2025c), with the performance comparable to that of Claude 3.5 Sonnet (49% by the 241022 version). The pass@10 of our SWE-Agent adapted model is 74.0% and surpasses the pass@30 of our model under Agentless (73.8%), proving the higher potential for the SWE-Agent framework.

## 4.2    SKILL TRANSFER AND GENERALIZATION

The results shown in Section 4.1 demonstrate that Kimi-Dev, a model with extensive Agentless training, could be adapted to end-to-end SWE-Agents with lightweight supervised finetuning. As the Agentless training recipe consists of mid-training, cold-start (SFT) and RL, we explore the contribution of each part in the recipe to the SWE-Agent capability after adaptation.

To figure this out, we perform SWE-Agent SFT on the original Qwen2.5-72B (Base), the mid-trained model (MT), the model then activated with Agentless-formatted long CoT data (SFT), and the (Kimi-Dev) model after finishing RL training (RL). As we are treating the four models as the *prior* for SWE-Agents[1], and a good prior always demonstrates the ability of fast adaptation with a few shots (Finn et al., 2017; Brown et al., 2020), we also sweep the amount of SWE-Agent SFT data to measure the *efficiency* of each prior in SWE-Agent adaptation.

Specifically, we randomly shuffle the 5,016 SWE-Agent trajectories and construct nested subsets of sizes 100, 200, 500, 1,000, and 2,000, where each smaller subset is contained within the larger ones. In addition, we prepend two extreme baselines: (i) zero-shot, where the prior model is directly evaluated under the SWE-Agent framework without finetuning, and (ii) one-step gradient descent, where the model is updated with a single gradient step using the 100-trajectory subset. This yields a range of SFT token budgets spanning $\{0, 2^{21}, 2^{23}, 2^{24}, 1.1 \times 2^{25}, 1.1 \times 2^{26}, 1.1 \times 2^{27}, 1.5 \times 2^{28}\}$. After these lightweight SFT experiments, we evaluate performance in terms of pass@$\{1,2,3\}$ under the SWE-Agent framework, with evaluations for pass@1 conducted at temperature 0, and those for pass@2 and pass@3 at temperature 1.0.

Figure 5 presents the SWE-Agent performances of each prior (Base, MT, SFT, RL) after being fine-tuned with different amounts of agentic trajectories. We have the following observations:

---

[1]We slightly abuse the term "prior" to refer to a model to be finetuned with SWE-Agent trajectories in the following analysis.

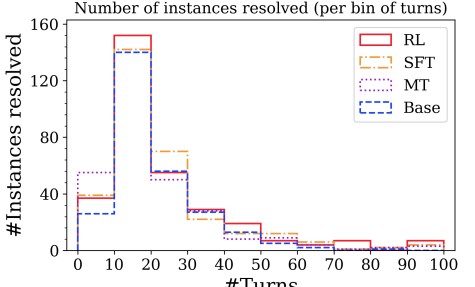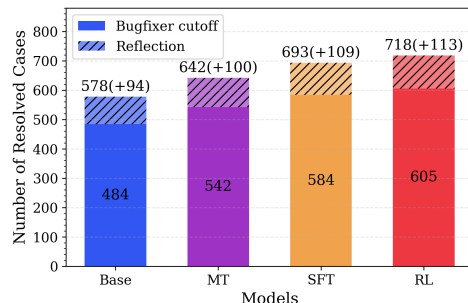

Figure 6: Left: Performance of the four priors under turn limits after SWE-Agent adaptation. Right: The characterization of the BugFixer and the reflection skills for each prior by counting the resolved cases of the 3 runs at Stage-3 cutoff moment, and comparing those with the final success cases.

1. The RL prior outperforms all the other models in nearly all the SWE-Agent SFT settings. This demonstrates that the Agentless training recipe indeed strengthens the prior in terms of SWE-Agent adaptation. For example, To achieve the top pass@1 performance of the Base prior, the RL prior needs only $2^{23}$ SWE-Agent SFT tokens, whereas the Base prior consumes $1.5 \times 2^{28}$ tokens.

2. The MT prior is lagged behind the SFT and the RL ones in extremely data-scarce settings (zero-shot (0) and one-step gradient descent ($2^{21}$) ), but quickly becomes on par with them after 200 trajectories ($2^{24}$) are available for finetuning. This indicates that adaptation efficiency remains comparable after the prior is strengthened through Agentless mid-training.

3. The performance of the SFT prior is mostly similar to the RL one except for two cases: (i) The SFT prior outperforms the RL one under the zero-shot setting. This is reasonable, as the RL prior might overfit to the Agentless input-output format, while the SFT prior suffers less from this. (ii) The SFT prior exhibits a significant degradation with 200 SWE-Agent trajectories ($2^{24}$). A potential reason could be that the 200 trajectories collapse onto a single data mode, leading the SFT prior to overfit through memorization (Chu et al., 2025); the RL prior instead embeds stronger transferable skills and thus generalizes better.

**From long CoT to extended multi-turn interactions.** We hypothesize that reflective behaviors cultivated through long chain-of-thought reasoning may transfer to settings requiring extended multi-turn interactions. To examine this, we evaluate the four priors (Base, MT, SFT, and RL) by finetuning on the 5,016 trajectories and test on SWE-bench Verified, under varying turn limits with pass@3 as the metric (Figure 6, left). The distinct interaction-length profiles show supportive evidence: the RL prior, after finetuning, continues to make progress beyond 70 turns, while the SFT, mid-trained, and raw models show diminishing returns around 70, 60, and 50 turns, respectively.

We further evaluate the efficacy of the Agentless skill priors (**BugFixer** and **reflection**) in the SWE-Agent adapted model. For **BugFixer**, given that the SWE-Agent may autonomously reflect between the five stages, we examine the moment in each trajectory when the bug fix of the third stage is *initially* completed, and the test rerun of the fourth stage has not yet been entered. Heuristically, when the SWE-Agent just completes the third stage, it has not yet obtained the execution feedback from the fourth stage, and thus has not further reflected based on the execution information or refined the bug fix. We therefore calculate the success rate of direct submission at this cutoff moment, which reflects the capability of the BugFixer skill. Regarding **reflection**, we further compare the performance at the cutoff point with the performance after full completion for each problem. The increment in the number of successful problems is used to reflect the capability of the reflection skill.

We use kimi-k2-0711-preview (Kimi et al., 2025) to annotate the SWE-Agent trajectories, identifying the stage to which each turn belongs. Figure 6 (right) demonstrates that both skills are strengthened through each stage of the Agentless training recipe: For the BugFixer skill, the cutoff performance at Stage-3 within the SWE-Agent interaction trajectories of the four adapted models shows consistent improvement, ranging from 484 cases resolved by the Base prior to 605 cases by the RL prior, as measured by the number of successful resolutions within three passes. For the reflection skill, examining the performance gains from Stage-3 to the end of the trajectories reveals a similar trend, with improvements increasing from +94 under the Base prior to +113 under

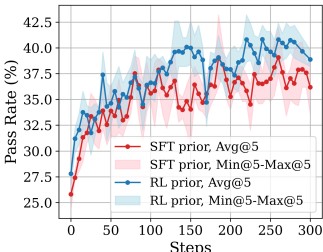 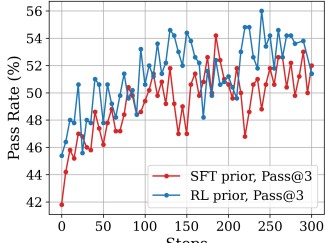 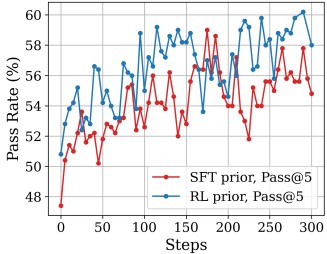

Figure 7: Comparison between the **SFT Prior** and the **RL Prior** when further applied with end-to-end SWE-Agent RL. Left: Pass@1 averaged from 5 runs. Middle: Pass@3. Right: Pass@5. The two priors are activated with the same $2^{21}$ SWE-Agent SFT tokens (the second column in Figure 5). After end-to-end RL, the RL prior slightly outperforms the SFT prior in all the Pass@1, Pass@3, and Pass@5 settings, which agrees with their SWE-Agent SFT performance comparison in Figure 5.

the RL prior. Taken together, the adapted model from the RL prior achieves the strongest overall performance across both skills. It should be noted that our analysis of the reflection skill remains coarse-grained, since the measured performance gains between the two checkpoints capture not only agentic reflection and redo behaviors, but also the intermediate test-writing process performed by the SWE-Agent. A more fine-grained evaluation that isolates the TestWriter skill prior is left for future work. The prompt for SWE-Agent stage annotation, extended qualitative studies, as well as additional discussions for skill transfer and generalization, are covered in Appendix G.

**End-to-end SWE-Agent RL for prior comparison.** To further validate the effectiveness of the priors baked by the Agentless training recipes, we employ end-to-end SWE-Agent RL (Luo et al., 2025) with the cold-started priors as the initial models. To maximally alleviate the effect from the patterns of proprietary models within the SWE-Smith trajectories, we leverage the setting with $2^{21}$ SWE-Agent SFT tokens, the second column in Figure 5, where a single step of gradient decent takes place on top of each prior. Under the minimal cold-start setup, end-to-end RL reveals the potential of each prior beyond taking the shortcut of imitation (Gudibande et al., 2024; Chu et al., 2025).

To run the end-to-end RL training for prior comparison, we use the SWE-Gym (Pan et al., 2024) and the SWE-bench-extra (Badertdinov et al., 2024a) subsets as the training set. Similarly to the Agentless RL recipe, we first use each initial model to filter out the problems with Pass@8 = 0. For the model with the MT prior, 260 out of 6,202 problems remain; for the models with the SFT prior and the RL prior, a total of 2,062 from the 6,202 problems are kept. In all end-to-end RL runs, we use the outcome reward only, and the same policy gradient algorithm in Sec. 3.3 without KL or entropy regularization for optimization, with batch size as 256. The results are shown as follows:

For the model with MT prior, the pass@1 performance quickly deteriorates to less than 2% after 10 end-to-end RL steps. The potential reason for this could be the lack of available problems to be trained with, reflecting the inferiority of the prior. For the models with the SFT prior and the RL prior, the RL runs last for 300 steps, and we plot the performance comparison in Figure 7. According to Figure 7, the model with the RL prior demonstrates slightly higher scores of Pass@1, Pass@3, and Pass@5 over the model with the SFT prior. While the phenomenon agrees with the performance comparison under SWE-Agent SFT shown in Figure 5, we observe that the patterns in the interaction trajectories of the models incentivized by end-to-end SWE-Agent RL significantly differ from the patterns of the proprietary models (detailed in Appendix G.3). These results reveal that the Agentless training recipe curates strong priors for end-to-end learning under SWE-Agent frameworks with the minimal supervision of proprietary end-to-end trajectories. We leave the exploration of more advanced agentic RL techniques for further improvement as future work.

## 5 CONCLUSION AND FUTURE WORK

In this work, we reframed Agentless and agentic paradigms for automated software engineering as complementary rather than competing. By introducing Kimi-Dev, we demonstrated that structured Agentless training can induce transferable skill priors, including bug localization, code repair, and self-reflection. As a result, Kimi-Dev not only achieves SoTA results on SWE-bench Verified among the workflow-based approaches, but enables efficient SWE-Agent adaptation as well. These findings establish a novel path toward building more generalizable coding agents through staged training.

## ACKNOWLEDGEMENTS

This research was conducted with the support of Fundamental and Interdisciplinary Disciplines Breakthrough Plan of the Ministry of Education of China (No. JYB2025XDXM101), the Shanghai Qi Zhi Institute & Spirit AI Innovation Program, and the Tsinghua University Dushi Program. Funding and support for this work were also provided by the Tsinghua University - Keystone Electrical (Zhejiang) Co.,Ltd Joint Research Center for Embodied Multimodal Artificial Intelligence (JCEMAI). Additionally, we would like to extend our thanks to the Xiongan AI Institute.

We thank Yuzhi Wang, Xinyu Zhou, Guokun Lai, Yulun Du, Fang Li, Hao Ding, Dehao Zhang, Enming Yuan, Dikang Du, and Jiacheng You for their valuable suggestions. We also appreciate the members of the infrastructure team at Moonshot AI for their timely support during the project.

## ETHICS AND REPRODUCIBILITY STATEMENTS

This work obeys the Code of Ethics required by the ICLR conference. The study does not involve human subjects or animal experimentation. The personally identifiable information from raw data is excluded for privacy consideration (see the mid-training data recipe detailed in Appendix A). Beyond the scope of this work, we strongly advocate for the community to advance systematic research on agent safety, thereby ensuring responsible progress in this area.

For all of the experiments, we have covered the detailed setups and discussions in the appendices: mid-training for Agentless in Appendix A, details of the used dockers in Appendix B, Agentless RL in Appendix C, agent infrastructure in Appendix D, case studies under Agentless in Appendix E, preliminary findings about emergent test-time parallel scaling in Appendix F, and extended analysis for SWE-Agents in Appendix G.

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

APPENDIX

## A  DETAILS OF MID-TRAINING

We curate a mid-training data recipe with a focus on enhancing SWE capabilities. Central to this effort is the collection of pull request (PR) data from GitHub, which provides extensive coverage of real-world bug fixes, feature requests, and code enhancements. To ensure data quality, we apply two filters: (i) we only retain repositories that have accumulated at least five GitHub stars, thereby excluding sparsely maintained projects with limited community engagement; and (ii) we remove any repositories overlapping with the SWE-bench benchmark (Jimenez et al., 2023) to prevent potential data leakage. For each candidate repository, we query the GitHub API for all PRs with the state MERGED, while discarding those abandoned, superseded, or left under review. To preserve more context information, we also snapshot the entire codebase at the base commit before the first code change in the PR.

After data crawling, we incorporate two complementary forms for the natural code change data: (i) **natural diff patches** and (ii) **PR commit packs**. A natural diff patch consolidates all commits in a PR into the final code difference, typically expressed as SEARCH–REPLACE blocks. This format aligns with the Agentless paradigm, in which the model must directly output the final patch. In contrast, a commit pack captures the sequence of human-authored commits within a PR, where each commit message (textual reasoning) is paired with the corresponding code modification (action). This structure closely parallels the SWE-Agent setting, where intermediate reasoning steps are interleaved with actions. However, the distinction of the utilities for the two types of data is not absolute: commit messages in a PR commit pack can still inform the model's knowledge and indirectly strengthen its reasoning ability in the Agentless setting.

**Natural diff patches**. The natural diff patches used in the mid-training data recipe are processed with the following rules:

- Incorporate the agentless prompt template (see Prompts 1,2,3,4; These four prompt templates are also used in the later stages, including cold-start, RL, and test-time self-play), and apply a loss mask to the prompt part. For the localization prompt, the response is set as the files modified in the ground-truth diff patch.

- If a related issue to the PR exists, use its content of the related issue; otherwise, use the PR title as the surrogate of the issue content.

- If a related issue to the PR exists, prepend the issue discussion at the beginning of the output in the code edit response. We aim to strengthen the model's capability of code edit reasoning by leveraging the discussion contents.

- Discard PRs that include modifications to files other than {.py, .md, .rst}.

- For PRs containing {.md, .rst} file modifications, retain only the Python diffs and rewrite them into SEARCH–REPLACE blocks.

- Remove PRs involving file additions or deletions.

- For the code edits with only line insertions or deletions, preserve the original Git diff hunks as the SEARCH content in the SEARCH–REPLACE blocks.

- Ensure that no more than three Python files are modified per PR.

- Apply a filtering script to exclude PRs with non-{.py, .md, .rst} modifications, or PRs modifying more than three Python files.

- Further exclude PRs containing more than five SEARCH–REPLACE blocks.

A total of ∼50B tokens for natural diff patches are obtained after applying these filtering rules.

```
Please look through the following GitHub problem description and
    Repository structure and provide a list of files that one would need
     to edit to fix the problem.

### GitHub Problem Description ###
{related issue / PR title content}
```

```
###

### Repository Structure ###
{file structure induced by the repo snapshot}

###

Please only provide the full path and return at most 5 files.
The returned files should be separated by new lines ordered by most to
    least important and wrapped with ```
For example:
```
file1.py
file2.py
```
```

Listing 1: Agentless prompt template: Localization for BugFixer.

```
Please look through the following GitHub problem description and
    Repository structure and provide a list of test files that should be
     run after applying the patch to fix the issue.

### GitHub Problem Description ###
{related issue / PR title content}

###

### Repository Structure ###
{file structure induced by the repo snapshot}

###

Please only provide the full path and return at most 5 files.
The returned files should be separated by new lines ordered by most to
    least important and wrapped with ```
For example:
```
file1.py
file2.py
```
```

Listing 2: Agentless prompt template: Localization for TestWriter.

```
We are currently solving the following issue within our repository.
    Here is the issue text:
--- BEGIN ISSUE ---
{related issue / PR title content}
--- END ISSUE ---

Below are some code segments, each from a relevant file. One or more of
     these files may contain bugs.

--- BEGIN FILE ---
```
### {filename1}
{content of filename1}

### {filename2}
{content of filename2}
{...}
```

```
```
--- END FILE ---

Please first localize the bug based on the issue statement, and then
    generate *SEARCH/REPLACE* edits to fix the issue.

Every *SEARCH/REPLACE* edit must use this format:
1. The file path
2. The start of search block: <<<<<<< SEARCH
3. A contiguous chunk of lines to search for in the existing source
    code
4. The dividing line: =======
5. The lines to replace into the source code
6. The end of the replace block: >>>>>>> REPLACE

Here is an example:

```python
### mathweb/flask/app.py
<<<<<<< SEARCH
from flask import Flask
=======
import math
from flask import Flask
>>>>>>> REPLACE
```

Please note that the *SEARCH/REPLACE* edit REQUIRES PROPER INDENTATION.
     If you would like to add the line '        print(x)', you must
    fully write that out, with all those spaces before the code!
Wrap the *SEARCH/REPLACE* edit in blocks ```python...```.
```

Listing 3: Agentless prompt template: Code edit for BugFixer.

```
We are currently solving the following issue within our repository.
    Here is the issue text:
--- BEGIN ISSUE ---
{related issue / PR title content}
--- END ISSUE ---

Below are some code segments, each from a relevant test file. One or
    more of these files may be added some new tests which can reproduce
    the issue.

--- BEGIN FILE ---
```
### {filename1}
{content of filename1}

### {filename2}
{content of filename2}
{...}

```
--- END FILE ---

Please first localize some possible locations in those test files
    within the repo, and then generate *SEARCH/REPLACE* edit updates to
    the **test** files in the repo, so that the erroneous scenario
    described in the problem is reproduced.

Every *SEARCH/REPLACE* edit must use this format:
1. The file path
```

```
2. The start of search block: <<<<<<< SEARCH
3. A contiguous chunk of lines to search for in the existing source
    code
4. The dividing line: =======
5. The lines to replace into the source code
6. The end of the replace block: >>>>>>> REPLACE

Here is an example:

```python
### mathweb/flask/app.py
<<<<<<< SEARCH
from flask import Flask
=======
import math
from flask import Flask

def test__rules__std_L060_raised() -> None:
    try:
        sql = "SELECT   IFNULL(NULL, 100),
            NVL(NULL,100);"
        result = lint(sql, rules=["L060"])
        assert len(result) == 2
    except:
        print("Other issues")
        return

    try:
        assert result[0]["description"] == "Use 'COALESCE' instead of '
            IFNULL'."
        assert result[1]["description"] == "Use 'COALESCE' instead of '
            NVL'."
        print("Issue resolved")
    except AssertionError:
        print("Issue reproduced")
        return

    return
>>>>>>> REPLACE
```

Please note that the *SEARCH/REPLACE* edit REQUIRES PROPER INDENTATION.
    If you would like to add the line '        print(x)', you must
    fully write that out, with all those spaces before the code!
Wrap the *SEARCH/REPLACE* edit in blocks ```python...```.
```

Listing 4: Agentless prompt template: Code edit for TestWriter.

**PR commit packs.** The PR commit packs used in the mid-training data recipe are processed with the following rules:

- Discard PRs that include modifications to files other than {.py, .md, .rst}.

- For {.md, .rst} file modifications, retain the "diff –git" signature but remove the actual content changes.

- Ensure that each PR modifies at most five Python files (with at least one required). PRs exceeding this limit are discarded.

- Apply a filtering script to exclude PRs containing non-{.py, .md, .rst} file modifications or those modifying more than five Python files.

- Filter out all of the developer signatures and GitHub IDs for ethics considerations.

A total of ∼20B tokens for PR commit packs are obtained after applying these filtering rules.

In addition, we incorporate synthetic data to further enhance both the reasoning and agentic capabilities of the model. A key observation is that the ground-truth reward for the localization stage in the Agentless setting can be derived directly from the diff patch, since the set of files requiring modification is explicitly indicated.

**Synthetic reasoning data.** To improve reasoning quality, we perform a lightweight SFT of the Qwen-2.5-72B-Instruct model on 2,000 R1 trajectories. The resulting model is then used to generate large-scale rollouts for the localization stage of both BugFixer and TestWriter. We retain only the rollouts that achieve exactly correct file localizations. This procedure yields approximately ∼10B tokens of reasoning-intensive data dedicated to Agentless localization in the mid-training recipe.

**Synthetic agentic interactions.** To strengthen agentic capabilities, we simulate agent–environment interactions with a custom tool set designed to mimic file-system operations without execution. This design is motivated by practical constraints: while repository snapshots from GitHub are available, not all snapshots are equipped with an executable Docker environment. As a result, shell commands are disabled. Instead, we introduce synthetic tools that allow the agent to view file contents and perform keyword-based search for localization, which effectively reproduces the first stage of Agentless but in an agentic manner. The specification of this tool set is covered in the system prompt, which is then used to elicit agentic interaction rollouts from the Qwen-2.5-72B-Instruct model. The complete system prompt is provided in Prompt 5. We apply a loss mask only to the system prompt, and enable the model to simultaneously learn both actions and observations along the trajectory, inspired by Yang et al. (2024b). This approach integrates both policy and world modeling into mid training.

```
Your job is to look through the given GitHub problem description and
    Repository structure, and edit updates to the files in the repo to
    resolve the problem.
The job is divided into two stages:
+ In Stage 1, you should localize the files the files that you would
    need to edit to fix the problem.
+ In Stage 2, you should edit the updates to the repo.
Let's begin from Stage 1 to localize the bugs:

In Stage 1, besides reading the provided Repository structure, you can
    use the following skills for exploration. The skills are to be
    called in an environment wrapped by <execute> and </execute>, listed
     in the form of python functions as below:

open_file(path: str, is_all | None = False, line_number: int | None =
    1, context_lines: int | None = 100) -> None:
    Opens the file at the given path in the editor for exploration.
    By default, only the first 100 lines of the file are displayed. To
        open the entire file, set 'is_all' to 'True'.
    The 'context_lines' parameter determines the maximum number of
        lines to be displayed, with a cap of 100 lines. Use 'scroll_up'
        and 'scroll_down' to view more content up or down.
    If a 'line_number' is provided, the window will be moved to include
         that line.
    Note: When 'is_all' is set to 'True', the 'line_number' and '
        context_lines' parameters will not take effect, as the entire
        file will be opened and displayed without any line-specific
        focus or context limitation.
    Args:
    path: str: The path to the file to open. the full path of the
        filename should be provided.
    is_all: bool | None = False: If set to 'True', the entire file will
         be opened. Defaults to 'False'.
    line_number: int | None = 1: The line number to move to. Defaults
        to 1.
    context_lines: int | None = 100: Only shows this number of lines in
         the context window (usually from line 1), with line_number as
        the center (if possible). Defaults to 100.

goto_line(line_number: int) -> None:
    Moves the window to show the specified line number.
```

```
    Args:
    line_number: int: The line number to move to.

goto_class_or_func(class_or_func_name: str) -> None:
    Moves the window to show the specified class or function in the
        current open file.
    Args:
    class_or_func_name: str: The name of the given class, function, or
        method in a class to move to.

scroll_down() -> None:
    Moves the window down by 100 lines.
    Args:
    None

scroll_up() -> None:
    Moves the window up by 100 lines.
    Args:
    None

search_dir(search_term: str, dir_path: str | None) -> None:
    Searches for search_term in all files in dir. If dir is not
        provided, searches in the entire repository. Filename, fine-
        grained line number, and the relative class or function it is
        located in (if applied) will be shown for each found position.
    Args:
    search_term: str: The term to search for.
    dir_path: str: The path to the directory to search. Should be full
        path filename.

search_file(search_term: str, file_path: str | None = None) -> None:
    Searches for search_term in file. If file is not provided, searches
         in the current open file. Filename, fine-grained line number,
        and the relative class or function it is located in (if applied)
         will be shown for each found position.
    Args:
    search_term: str: The term to search for.
    file_path: str | None: The path to the file to search. Should be
        full path filename if provided.

find_file(file_name: str, dir_path: str | None) -> None:
    Finds all files with the given name in the specified directory. If
        dir is not provided, find in the entire repository.
    Args:
    file_name: str: The name of the file to find.
    dir_path: str: The path to the directory to search.

str_replace(path: str, old_str, new_str)
old_str=[the old content to be replaced]
new_str=[the new content after replacement]
-> None:
    Replace the old content (old_str) in the file at the given path
        with the new content (new_str). This is the skill that you will
        be using to edit the updates.
    Args:
    path: str: The path to the file to be updated. The full path of the
         filename should be provided.
    old_str: str: The old content to be replaced. Note that this
        argument should be written in a new line starting with "old_str
        =", and the string content should not be quoted.
    new_str: str: The new content after replacement. Note that this
        argument should be written in a new line starting with "new_str
        =", and the string content should not be quoted.

    Example:
```

```
    Assuming a call is shown as follows:
```
str_replace("filename.py", old_str, new_str)
old_str=    a

new_str=    b
c
```
    Then it will function as replacing the '    a\n' string with the '
         b\nc ' string in the 'filename.py' file.

insert(path: str, insert_line: int, new_str)
new_str=[the new content to be inserted]
-> None:
    Insert the new content (new_str) in the file at the given path.
         When you want to add an entirely new class/function to the file,
          it would be better to use this method.
    Args:
    path: str: The path to the file to be updated. The full path of the
         filename should be provided.
    insert_line: int: The Line number below which the new content is to
         be added. This Line number should be within the range of lines
         of the file: [0, Lines_of_the_File]. Specifically, when
         insert_line = 0, the added content starts from the top of the
         file.
    new_str: str: The new content to be inserted. Note that this
         argument should be written in a new line starting with "new_str
         =", and the string content should not be quoted.

    Example:
    Assuming a call is shown as follows:
```
insert("test_filename.py", 5, new_str)
new_str=    def test_add():
        assert add(1, 2) == 3
```
    Then it will function as inserting the string '    def test_add():\
        n        assert add(1, 2) == 3' below the Line 5 of the '
        test_filename.py' file.

stop() -> None:
    Terminate the editing process.
    Args:
    None

NOTE:
Responses should be concise.
When exploring, you should attempt fewer things at a time: Include ONLY
     ONE <execute> per response, and use a SINGLE skill listed above
    within the <execute> environment. DO NOT use other python functions,
     as the environment does not support them.
You should first reason in the verbal form, then use a skill with <
    execute> and </execute>.
You should avoid apologies and thanks in the responses.

When you finish exploring and analyzing with the provided skills,
    please return at most 3 files with the full path only. Each full
    path should be placed in a single line, INSTEAD OF BROKEN WITH
    MULTIPLE LINES.
The returned files should be separated by new lines ordered by most to
    least important, wrapped with ``` and NOTHING ELSE.
An example for a full output:
```
full_path_to_file1.py
```

```
full_path_to_file2.py
```

Now Let's start!

### GitHub Problem Description ###

{issue content}

### Repository Structure ###

{file structure}

###
```

Listing 5: A non-execution set of tools empowering the simulation of agentic interaction trajectories.

After completing the initial localization stage, the agent is guided into the code-editing phase through a follow-up instruction: "Now let's move on to Stage 2 and edit the updates. Remember, you can still decide at any point whether a file actually requires modification." We retain partial rollouts from Stage 1, provided that the localization results include at least one correct file.

In Stage 2, we first simulate the agent's interaction by allowing it to open incorrectly localized files, and we artificially inject agentic reasoning patterns such as "I realize that I do not need to modify this file" after inspecting the file content. This procedure is designed to strengthen the self-reflection ability of the agent by exposing it to false-positive contexts regarding the issue to be solved.

Subsequently, we transcribe the ground-truth PR commit pack into trajectory form: each commit message is treated as the agent's reasoning step, and each code update is represented as the corresponding action, expressed through the "str_replace" or "insert" tools. These interactions are appended to the trajectory, followed by a terminating "stop" call. Due to storage constraints on repository snapshots, this trajectory simulation is applied to only a subset of PRs. Overall, this process contributes approximately $\sim$10B tokens of agentic interaction data to the mid-training recipe. Future directions for scaling this component in the data recipe include leveraging the idea of environment scaling (Yang et al., 2025c).

**Training.** We perform mid-training using a standard next token prediction approach, initialized from the Qwen2.5-72B-Base (Qwen et al., 2024) model. We upsample the synthetic part of the data by a factor of 4 during mid-training, inspired by the practice in Grattafiori et al. (2024); Qwen et al. (2024); Gu et al. (2025). A global batch size of 256 with a maximum sequence length of 32K tokens is used, optimizing for long-context capabilities necessary for real-world software engineering tasks. The learning rate is set to 2e-5, with a cosine decay schedule and a minimum learning rate of 2e-6. The warm-up phase covers over approximately 3 billion tokens, followed by learning rate decay until approximately 150 billion tokens are processed.

## B  DOCKER ENVIRONMENTS

Table 3: The sources of the docker environments used in the development of Kimi-Dev.

| Dataset Name | Dataset Link | Number of Dockers |
|---|---|---|
| SWE-Gym (Pan et al. (2024)) | `https://huggingface.co/ datasets/SWE-Gym/ SWE-Gym/` | 2,356 |
| SWE-bench-extra (Badertdinov et al. (2024a)) | `https://huggingface.co/ datasets/nebius/ SWE-bench-extra/` | 3,846 |
| R2E-Gym-Lite (Jain et al. (2025)) | `https://huggingface.co/ datasets/R2E-Gym/ R2E-Gym-Lite` | 3,671 |

**Docker environment construction**. To validate non-ground-truth patches generated by model roll-outs and expand our dataset, we required executable Docker environments. We combined publicly available datasets with custom-configured Docker environments (see Table. 3). Among them, SWE-Gym and R2E-Gym-Lite open-source their dockers that we can directly use. For datasets lacking Docker support (SWE-Bench-Extra), we implemented an automated configuration method:

1. Initialize a Docker environment with fixed dependencies.
2. Select Python version based on commit year.
3. Install dependencies via `requirements.txt` and "`pip install -e .`".
4. Resolve `ModuleNotFound` errors during test execution.
5. Validate success if a `FAIL_TO_PASS` test transitions from failing (without `gt_patch`) to passing (with `gt_patch`).

Out of 6.38k SWE-bench-extra instances, 3,846 environments are successfully constructed and subsequently used for cold-start and RL training.

## C  MORE DETAILS OF RL TRAINING

### C.1  PROMPT SET SELECTION

In the main text, we introduce the adaptive prompt selection method for RL training. Specifically, we construct an initial prompt set of 1,200 problems by selecting those with pass@16 > 0 from SWE-Gym (Pan et al., 2024), SWE-bench-extra (Badertdinov et al., 2025), and R2E-gym (Jain et al., 2025). Then, every 100 training steps, we expand the prompt set by adding 500 new problems. These additional problems are randomly sampled and filtered from the pool of problems for which the current model has pass@16 = 0, thereby progressively increasing the difficulty and forming a proper curriculum.

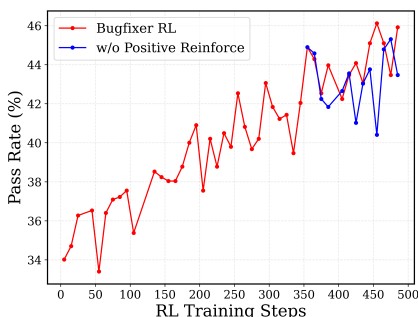

Figure 8: Ablation of positive example reinforcement during 72B Bugfixer RL.

### C.2  RL EXPERIMENT ABLATION

Figure 9 shows the performance of the Qwen2.5-14B model in RL experiments, where both the BugFixer and the TestWriter exhibit clear scaling law behavior.

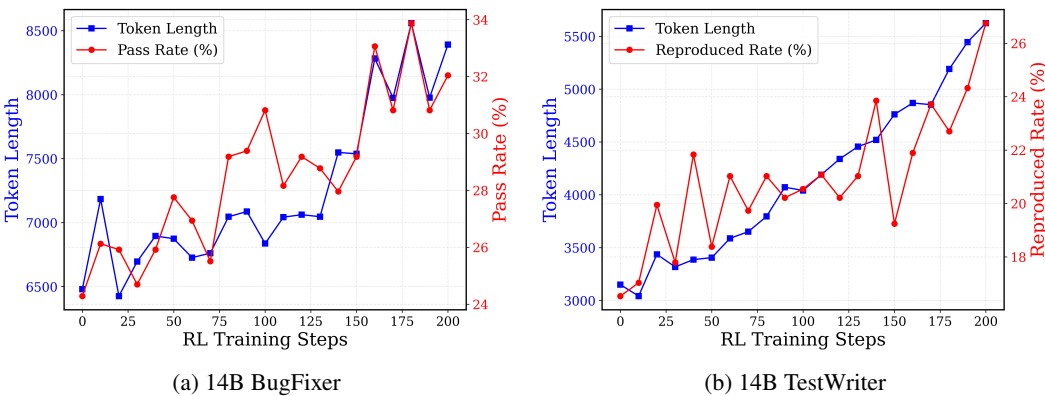

(a) 14B BugFixer                    (b) 14B TestWriter

Figure 9: RL scaling experiments on Qwen2.5-14B model.

Furthermore, Figure 8 illustrates the effect of our proposed positive example reinforcement. We incorporated this improvement in the later stage of training and found that the RL of BugFixer achieved better performance compared to not using this trick. The main reason is that positive

examples can reinforce the correct reasoning patterns, enabling the model to converge more quickly, especially when its exploration ability diminishes in the later stages. It is noticed that a similar trick was also applied in the development of the Seed1.5-thinking model (Seed et al., 2025).

## D  AGENT INFRASTRUCTURE

We present a unified sandbox architecture designed for hybrid cloud cluster environments, leveraging Kubernetes orchestration to provide researchers with a flexible, secure, and highly scalable code execution platform. The architecture is accessible through a Python SDK and implements an ephemeral, use-and-destroy paradigm that ensures resource efficiency and security isolation. The system demonstrates exceptional scalability, supporting over 10,000 concurrent sandbox instances while maintaining performance stability. This infrastructure serves as a foundational platform for diverse computational scenarios, including automated code evaluation and RL experiments.

The system employs a sidecar container pattern to efficiently orchestrate and manage a heterogeneous collection of over 25,000 docker images sourced from multiple datasets. This architectural choice enables optimal resource utilization while maintaining isolation between different execution environments. Additionally, we have developed a proprietary Software Engineering (SWE) image-building pipeline that continuously expands the repository of supported images, thereby enhancing the system's adaptability to diverse computational workloads and research requirements.

## E  CASE STUDIES UNDER AGENTLESS

### E.1  BUGFIXER TRACES

We provide a Bugfixer's response trace during the code repair stage (see Listing 6). From this, we can observe that the initial code given was `return self[i, j - pos - other.cols]`, but the model quickly realized its mistake and corrected it to `return self[i, j - other.cols]`. Although some repetitive self-reflections followed, the increasing response length indeed improved answer accuracy and robustness in our experiments.

```
<think>
Okay, let's tackle this problem. The user reported an issue where the
    Matrix.col_insert() method isn't working correctly. The example
    given shows that after inserting a matrix V into M at position 3,
    the resulting matrix has the inserted columns shifted incorrectly.
    The expected result would have the inserted columns at positions 3
    and 4, but the actual output shows them starting at position 3, but
    the original matrix's columns after the insert are shifted upwards
    instead of remaining in their original positions.

First, I need to understand how col_insert works. Looking at the code
    provided, especially in the MatrixShaping class in sympy/matrices/
    common.py, the _eval_col_insert method is responsible for inserting
    columns. The current implementation shifts the existing columns to
    the right by the number of columns inserted. However, the bug might
    be in how the original matrix's columns are being shifted.

Let me look at the existing code for _eval_col_insert:

def _eval_col_insert(self, pos, other):
    cols = self.cols

    def entry(i, j):
        if j < pos:
            return self[i, j]
        elif pos <= j < pos + other.cols:
            return other[i, j - pos]
        return self[i, j - pos - other.cols]

    return self._new(self.rows, self.cols + other.cols,
```

```
                            lambda i, j: entry(i, j))
```

Wait, the problem here is in the last case. When j is greater than or
    equal to pos + other.cols, the code subtracts (pos + other.cols)
    from j. But that's incorrect because the original columns after the
    inserted ones should start at j = pos + other.cols. So for the
    original matrix's columns that come after the inserted ones, their
    new positions should be j + other.cols. But in the current code,
    when j is in the original matrix's columns (after the inserted ones)
    , the code is subtracting (pos + other.cols), which would
    incorrectly shift them.

Wait, let's think again. The original code for the entry function:

```
if j < pos:
    return self[i, j]  # original columns before pos
elif pos <= j < pos + other.cols:
    return other[i, j - pos]  # inserted columns
return self[i, j - pos - other.cols]  # original columns after the
    inserted ones
```

Wait, no. Let's say the original matrix has cols = 6. We insert 2
    columns at pos=3. The new total cols is 6 + 2 = 8. The original
    columns after pos 3 (columns 3,4,5) should now be at positions
    5,6,7. But according to the code, when j is >=3 +2=5 (since pos +
    other.cols =3+2=5), then the code returns self[i, j -5 -2] = self[i,
     j -7]. Wait, that's not right. For example, if j=5 (the new column
    5), then j - pos - other.cols =5 -3 -2=0. So the original column 0
    would be mapped to the new column 5. That's definitely wrong.

Ah, here's the mistake. The code is subtracting (pos + other.cols) from
     j, but that's incorrect. The original columns after the inserted
    ones should be shifted to the right by other.cols. So, the correct
    calculation should be j - other.cols. For example, original column 3
     becomes 3 +2=5, original column 4 becomes 6, etc.

So the line in the code:

```
return self[i, j - pos - other.cols]
```

should be:

```
return self[i, j - other.cols]
```

Because the inserted columns have taken up 'other.cols' positions
    starting at 'pos', so the original columns after 'pos' are shifted
    by 'other.cols' to the right. Therefore, to get the correct original
     column index, we subtract 'other.cols' from the current j.

So the fix is to change the subtraction from (pos + other.cols) to
    other.cols. Let me check:

Original code:

```
return self[i, j - pos - other.cols]
```

Corrected code:

```
return self[i, j - other.cols]
```

Wait, no. Wait, the original columns after the inserted block (which
    starts at pos and has other.cols columns) would start at j = pos +
    other.cols. So for those j values, the original column index would
    be j - (pos + other.cols). Wait, no. Let's think with numbers.

```
Original matrix has 6 columns. Insert 2 columns at position 3. The new
    matrix has 8 columns. The original columns 0,1,2 remain in positions
     0,1,2. The inserted 2 columns are at positions 3 and 4. The
    original columns 3,4,5 now become 5,6,7.

So for j in 5,6,7 (original columns 3,4,5), the mapping is:

5 $\rightarrow$ 3

6 $\rightarrow$ 4

7 $\rightarrow$ 5

So the formula should be original_column = j - inserted_columns. In
    this case, inserted_columns is other.cols. So the code should
    subtract other.cols from j when accessing the original matrix's
    columns after the inserted block.

Therefore, the line:

return self[i, j - pos - other.cols]

should be:

return self[i, j - other.cols]

Because pos is where the insertion starts, but the original columns
    after the inserted block are shifted by other.cols. Let's verify
    with the example:

pos =3, other.cols=2. For j=5 (new column 5), original column index is
    5 -2=3, which is correct. For j=6, 6-2=4; j=7, 7-2=5. That's correct
    .

So the error in the original code was subtracting (pos + other.cols)
    instead of other.cols. The pos is the starting position of the
    insertion, but after inserting other.cols columns, the original
    columns after the inserted block are shifted by other.cols.
    Therefore, the correct subtraction is other.cols.

Thus, the fix is to change the line in the _eval_col_insert method from
    :

return self[i, j - pos - other.cols]

to:

return self[i, j - other.cols]

Now, looking back at the code in the _eval_col_insert function:

The code for the else clause is:

return self[i, j - pos - other.cols]

Which is incorrect. Changing that to j - other.cols should fix the
    problem.

So the SEARCH block would look for the lines:

    def _eval_col_insert(self, pos, other):
        cols = self.cols

        def entry(i, j):
            if j < pos:
```

```
                    return self[i, j]
             elif pos <= j < pos + other.cols:
                    return other[i, j - pos]
             return self[i, j - pos - other.cols]

         return self._new(self.rows, self.cols + other.cols,
                           lambda i, j: entry(i, j))
```

And the REPLACE block would change the last part to:

```
             elif pos <= j < pos + other.cols:
                    return other[i, j - pos]
             return self[i, j - other.cols]
```

So the correction is in the return statement of the entry function. The
    original code subtracted (pos + other.cols) from j, but it should
    subtract other.cols instead.

Therefore, the correct SEARCH/REPLACE block would be:

```
### sympy/matrices/common.py
<<<<<<< SEARCH
    def _eval_col_insert(self, pos, other):
        cols = self.cols

        def entry(i, j):
             if j < pos:
                    return self[i, j]
             elif pos <= j < pos + other.cols:
                    return other[i, j - pos]
             return self[i, j - pos - other.cols]

        return self._new(self.rows, self.cols + other.cols,
                          lambda i, j: entry(i, j))
=======
    def _eval_col_insert(self, pos, other):
        def entry(i, j):
             if j < pos:
                    return self[i, j]
             elif pos <= j < pos + other.cols:
                    return other[i, j - pos]
             return self[i, j - other.cols]

        return self._new(self.rows, self.cols + other.cols,
                          lambda i, j: entry(i, j))
>>>>>>> REPLACE
```

This change corrects the column shifting calculation for the original
    matrix's columns after the inserted block. Instead of subtracting (
    pos + other.cols), we subtract other.cols, which properly shifts the
     original columns to the right by the number of inserted columns.
</think>

```python
### sympy/matrices/common.py
<<<<<<< SEARCH
    def _eval_col_insert(self, pos, other):
        cols = self.cols

        def entry(i, j):
             if j < pos:
                    return self[i, j]
             elif pos <= j < pos + other.cols:
                    return other[i, j - pos]
             return self[i, j - pos - other.cols]
```

```
        return self._new(self.rows, self.cols + other.cols,
                          lambda i, j: entry(i, j))
=======
    def _eval_col_insert(self, pos, other):
        def entry(i, j):
            if j < pos:
                return self[i, j]
            elif pos <= j < pos + other.cols:
                return other[i, j - pos]
            return self[i, j - other.cols]

        return self._new(self.rows, self.cols + other.cols,
                          lambda i, j: entry(i, j))
>>>>>>> REPLACE
```
```

Listing 6: Reasoning trajectory example for BugFixer.

## E.2 TESTWRITER FALSE POSITIVE CASES

During the RL training process, we observed that the Testwriter component exhibits some false
positives, which can result in tests that contain certain gaps. This might cause the test patches
generated by the model to be mistakenly considered as correct. Consequently, the performance
improvement of Testwriter does not accurately reflect the actual performance. In this section, we
provide two intuitive examples to illustrate this phenomenon, and leave further enhancement for
TestWriter as future work.

Listing 7 shows the ground-truth test patch for instance django_django-16877, with Listing
8 showing the test patch generated by theTestWriter. A detailed comparison reveals that the model-
generated test cases are not comprehensive: Two locations should have been tested, but only one
was checked, and the effect of mark_safe was not considered. In practice, this allowed the model-
predicted patch to pass the TestWriter reward check, but result in a false positive.

The second example instance is sympy_sympy-13974, with Listings 9 and 10 showing the
ground-truth and model-generated test patches. In this case, the coverage of the generated test is
also insufficient. The generated test does not account for the need to satisfy the associativity prop-
erty, e.g., assert tensor_product_simp(TP(A, B)**x) == TP(A**x, B**x). As
a consequence, it fails to serve as a reliable judge for BugFixer.

```
diff --git a/tests/template_tests/filter_tests/test_escapeseq.py b/
    tests/template_tests/filter_tests/test_escapeseq.py
new file mode 100644
--- /dev/null
+++ b/tests/template_tests/filter_tests/test_escapeseq.py
@@ -0,0 +1,59 @@
+from django.test import SimpleTestCase
+from django.utils.safestring import mark_safe
+
+from ..utils import setup
+
+
+class EscapeseqTests(SimpleTestCase):
+    """
+    The "escapeseq" filter works the same whether autoescape is on or
    off,
+    and has no effect on strings already marked as safe.
+    """
+
+    @setup(
+        {
+            "escapeseq_basic": (
+                '{{ a|escapeseq|join:", " }} -- {{ b|escapeseq|join:",
    " }}'
```

```
+                ),
+            }
+        )
+    def test_basic(self):
+        output = self.engine.render_to_string(
+            "escapeseq_basic",
+            {"a": ["x&y", "<p>"], "b": [mark_safe("x&y"), mark_safe("<
    p>")]},
+        )
+        self.assertEqual(output, "x&y, <p> -- x&y, <p>")
+
+    @setup(
+        {
+            "escapeseq_autoescape_off": (
+                '{% autoescape off %}{{ a|escapeseq|join:", " }}'
+                " -- "
+                '{{ b|escapeseq|join:", "}}{% endautoescape %}'
+            )
+        }
+    )
+    def test_autoescape_off(self):
+        output = self.engine.render_to_string(
+            "escapeseq_autoescape_off",
+            {"a": ["x&y", "<p>"], "b": [mark_safe("x&y"), mark_safe("<
    p>")]},
+        )
+        self.assertEqual(output, "x&y, <p> -- x&y, <p>")
+
+    @setup({"escapeseq_join": '{{ a|escapeseq|join:"
" }}'})
+    def test_chain_join(self):
+        output = self.engine.render_to_string("escapeseq_join", {"a":
    ["x&y", "<p>"]})
+        self.assertEqual(output, "x&y
<p>")
+
+    @setup(
+        {
+            "escapeseq_join_autoescape_off": (
+                '{% autoescape off %}{{ a|escapeseq|join:"
" }}{%
    endautoescape %}'
+            ),
+        }
+    )
+    def test_chain_join_autoescape_off(self):
+        output = self.engine.render_to_string(
+            "escapeseq_join_autoescape_off", {"a": ["x&y", "<p>"]}
+        )
+        self.assertEqual(output, "x&y
<p>")
```

Listing 7: GT test patch of `django__django-16877`.

```
diff --git a/tests/template_tests/filter_tests/test_safeseq.py b/tests/
    template_tests/filter_tests/test_safeseq.py
--- a/tests/template_tests/filter_tests/test_safeseq.py
+++ b/tests/template_tests/filter_tests/test_safeseq.py
@@ -20,3 +20,20 @@
     def test_safeseq02(self):
         output = self.engine.render_to_string("safeseq02", {"a": ["&",
             "<"]})
         self.assertEqual(output, "&, < -- &, <")
+
+    # New tests for escapeseq
+    @setup({"escapeseq01": '{{ a|escapeseq|join:", " }}'})
+    def test_escapeseq01(self):
+        output = self.engine.render_to_string("escapeseq01", {"a":
    ["&", "<"]})
```

```
+           self.assertEqual(output, "&, <")
+
+     @setup(
+         {
+             "escapeseq02": (
+                 '{% autoescape off %}{{ a|escapeseq|join:", " }}{%
+    endautoescape %}'
+             )
+         }
+     )
+     def test_escapeseq02(self):
+         output = self.engine.render_to_string("escapeseq02", {"a":
+    ["&", "<"]})
+         self.assertEqual(output, "&, <")
```

Listing 8: Model predicted test patch of `django__django-16877`.

```
diff --git a/sympy/physics/quantum/tests/test_tensorproduct.py b/sympy/
    physics/quantum/tests/test_tensorproduct.py
--- a/sympy/physics/quantum/tests/test_tensorproduct.py
+++ b/sympy/physics/quantum/tests/test_tensorproduct.py
@@ -10,7 +10,7 @@
 from sympy.physics.quantum.density import Density
 from sympy.core.trace import Tr

-A, B, C = symbols('A,B,C', commutative=False)
+A, B, C, D = symbols('A,B,C,D', commutative=False)
 x = symbols('x')

 mat1 = Matrix([[1, 2*I], [1 + I, 3]])
@@ -47,6 +47,11 @@ def test_tensor_product_commutator():

 def test_tensor_product_simp():
     assert tensor_product_simp(TP(A, B)*TP(B, C)) == TP(A*B, B*C)
+    # tests for Pow-expressions
+    assert tensor_product_simp(TP(A, B)**x) == TP(A**x, B**x)
+    assert tensor_product_simp(x*TP(A, B)**2) == x*TP(A**2,B**2)
+    assert tensor_product_simp(x*(TP(A, B)**2)*TP(C,D)) == x*TP(A**2*C
    ,B**2*D)
+    assert tensor_product_simp(TP(A,B)-TP(C,D)**x) == TP(A,B)-TP(C**x,
    D**x)

 def test_issue_5923():
```

Listing 9: GT test patch for `sympy__sympy-13974`.

```
diff --git a/sympy/physics/quantum/tests/test_tensorproduct.py b/sympy/
    physics/quantum/tests/test_tensorproduct.py
--- a/sympy/physics/quantum/tests/test_tensorproduct.py
+++ b/sympy/physics/quantum/tests/test_tensorproduct.py
@@ -47,6 +47,27 @@

 def test_tensor_product_simp():
     assert tensor_product_simp(TP(A, B)*TP(B, C)) == TP(A*B, B*C)
+
+    # Test tensor product powers
+    a = symbols('a', commutative=False)
+    t1 = TP(1, 1) * TP(1, 1)
+    t1_simp = tensor_product_simp(t1)
+    assert t1_simp == TP(1, 1), f"Expected 1x1, got {t1_simp}"
+
+    # Test with Pauli matrices
+    from sympy.physics.paulialgebra import Pauli
```

```
+    t2 = TP(1, Pauli(3)) * TP(1, Pauli(3))
+    t2_simp = tensor_product_simp(t2)
+    assert t2_simp == TP(1, 1), f"Expected 1x1, got {t2_simp}"
+
+    # Test substitution cases from issue
+    expr1 = TP(1, 1) * TP(1, a)
+    result1 = expr1.subs(a, 1)
+    assert tensor_product_simp(result1) == TP(1, 1), f"Substitution 1
   failed: {result1}"
+
+    expr2 = TP(1, Pauli(3)) * TP(1, a)
+    result2 = expr2.subs(a, Pauli(3))
+    assert tensor_product_simp(result2) == TP(1, 1), f"Substitution 2
   failed: {result2}"

 def test_issue_5923():
```

Listing 10: Model predicted test patch of `sympy__sympy-13974`.

## F  EMERGENT PARALLEL SCALING IN KIMI-DEV

Compared to the self-play method that leverages execution during test time, parallel scaling (Chen et al., 2025b; Pan et al., 2025) is a recently-introduced test-time scaling method that aggregates multiple independent rollouts for additional improvement. While previous works incentivize the ability of parallel scaling through training, or train a separate reward model (Pan et al., 2024; Jain et al., 2025) for patch selection, we observe that the parallel scaling effect emerges in our model *without any specific training*. Specifically, patch candidates from parallel rollouts of BugFixer are used to prompt the model to summarize their differences and compose a new bugfix patch. Listing 11 is the prompt template of parallel scaling. In

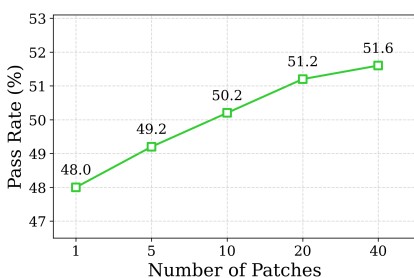

Figure 10: Emergent parallel scaling results on SWE-bench Verified.

the template, `promblem_statement` is the GitHub issue, and `trajs_content` represents the content of multiple patch candidates.

```
We are currently solving the following issue within our repository.
   Here is the issue text:
--- BEGIN ISSUE ---
{problem_statement}
--- END ISSUE ---

Below are some thinking trajectories, each from llm reasoning model.
   Only one trajetory is right.
--- BEGIN FILE ---
```
{trajs_content}
```
--- END FILE ---

Please first summary and analyze the key differences between the
   trajectories, and then generate *SEARCH/REPLACE* edits to fix the
   issue.

Every *SEARCH/REPLACE* edit must use this format:
1. The file path
2. The start of search block: <<<<<<< SEARCH
3. A contiguous chunk of lines to search for in the existing source
   code
4. The dividing line: =======
```

```
5. The lines to replace into the source code
6. The end of the replace block: >>>>>>> REPLACE

Here is an example:

```python
### mathweb/flask/app.py
<<<<<<< SEARCH
from flask import Flask
=======
import math
from flask import Flask
>>>>>>> REPLACE
```

Please note that the *SEARCH/REPLACE* edit REQUIRES PROPER INDENTATION.
    If you would like to add the line '          print(x)', you must
   fully write that out, with all those spaces before the code!
Wrap the *SEARCH/REPLACE* edit in blocks ```python...```.
The summary of the key differences between the trajectories should be
   in the thinking part.
```

Listing 11: The prompt template for parallel scaling.

The results in Figure 10 show that the performance of the parallel aggregation improves as the number of patch candidates in the prompt increases. The advantage of this scaling paradigm over majority voting lies in its ability to leverage the model's own capacity to analyze multiple candidate patches, thereby surpassing the simplistic approach of weighting answers merely by their frequency of occurrence. Listing 12 covers a full prompt example with 14 different input patch candidates. *Note: As Listing 12 is long, it is OK to jump to Listing 13 to directly observe the model behavior.*

```
We are currently solving the following issue within our repository.
    Here is the issue text:
--- BEGIN ISSUE ---
Data <@filename isn't relative to the YAML file
The [docs say](https://gabbi.readthedocs.io/en/latest/format.html#data)
    :

> If the value is a string that begins with <@ then the rest of the
    string is treated as the name of a file to be loaded from the same
    directory as the YAML file.

But I haven't found it works like this unless I cd into the directory
    containing the yaml file.

`_load_data_file` says:
```python
path = os.path.join(self.test_directory, os.path.basename(filename))
```
this does a few things:
- makes the path begin with `self.test_directory` (defaults to `.`, the
     current working directory)
- disguards any folders in the <@ `filename` path
- appends `filename`

This means, if I'm in `/`, and I have a test that says `data: <@cake.
    jpg`, I cannot run my tests as advised in the docs:
`gabbi-run -- /my/test.yaml /my/other.yaml`

`FileNotFoundError: [Errno 2] No such file or directory: './cake.jpg'`

So in our test running script, we have to start with `cd tests/`.
```

```
My preference, would be to make the path relative to the location of
    the yaml file. That way, I can run my tests from whatever directory.

Obviously this isn't possible when redirecting with `<` as the filename
     is never seen by gabbi. So I'm happy is discuss other ideas.

---

Furthermore I can't keep my test data in a child directory, unless I cd
     into that directory like:
```shell
cd yaml_tests/test_data
gabbi-run -- ../test_things.yaml
```

So for that reason, I'd like to allow directories like `<@test_data/
    filename.txt` to be included in the path.

---

Happy to write the patch, if we come up with an approach here.
--- END ISSUE ---

Below are some thinking trajectories, each from llm reasoning model.

--- BEGIN FILE ---
```
**Answer 1:**
```python
### gabbi/runner.py
<<<<<<< SEARCH
import argparse
from importlib import import_module
import sys
import unittest
=======
import argparse
from importlib import import_module
import os
import sys
import unittest
>>>>>>> REPLACE
```
```python
### gabbi/runner.py
<<<<<<< SEARCH
def run_suite(handle, handler_objects, host, port, prefix, force_ssl=
    False,
             failfast=False):
    """Run the tests from the YAML in handle."""
    data = utils.load_yaml(handle)
    if force_ssl:
        if 'defaults' in data:
            data['defaults']['ssl'] = True
        else:
            data['defaults'] = {'ssl': True}

    loader = unittest.defaultTestLoader
    test_suite = suitemaker.test_suite_from_dict(
        loader, 'input', data, '.', host, port, None, None, prefix=
            prefix,
        handlers=handler_objects)

    result = ConciseTestRunner(
```
```

```python
            verbosity=2, failfast=failfast).run(test_suite)
        return result.wasSuccessful()
=======
def run_suite(handle, handler_objects, host, port, prefix, force_ssl=
    False,
              failfast=False, test_directory='.'):
    """Run the tests from the YAML in handle."""
    data = utils.load_yaml(handle)
    if force_ssl:
        if 'defaults' in data:
            data['defaults']['ssl'] = True
        else:
            data['defaults'] = {'ssl': True}

    loader = unittest.defaultTestLoader
    test_suite = suitemaker.test_suite_from_dict(
        loader, 'input', data, test_directory, host, port, None, None,
        prefix=prefix, handlers=handler_objects)

    result = ConciseTestRunner(
        verbosity=2, failfast=failfast).run(test_suite)
    return result.wasSuccessful()
>>>>>>> REPLACE
```
```python
### gabbi/runner.py
<<<<<<< SEARCH
    else:
        for input_file in input_files:
            with open(input_file, 'r') as fh:
                success = run_suite(fh, handler_objects, host, port,
                                    prefix, force_ssl, failfast)
=======
    else:
        for input_file in input_files:
            test_directory = os.path.dirname(input_file) or '.'
            with open(input_file, 'r') as fh:
                success = run_suite(fh, handler_objects, host, port,
                                    prefix, force_ssl, failfast,
                                    test_directory=test_directory)
>>>>>>> REPLACE
```

**Answer 2:**
```python
### gabbi/runner.py
<<<<<<< SEARCH
def run_suite(handle, handler_objects, host, port, prefix, force_ssl=
    False,
              failfast=False):
=======
def run_suite(handle, handler_objects, host, port, prefix, force_ssl=
    False,
              failfast=False, test_directory='.'):
>>>>>>> REPLACE
```
```python
### gabbi/runner.py
<<<<<<< SEARCH
    test_suite = suitemaker.test_suite_from_dict(
        loader, 'input', data, '.', host, port, None, None, prefix=
            prefix,
        handlers=handler_objects)
=======
    test_suite = suitemaker.test_suite_from_dict(
```

```
            loader, 'input', data, test_directory, host, port, None, None,
                prefix=prefix,
            handlers=handler_objects)
>>>>>>> REPLACE
```
```python
### gabbi/runner.py
<<<<<<< SEARCH
                success = run_suite(fh, handler_objects, host, port,
                                    prefix, force_ssl, failfast)
=======
                test_directory = os.path.dirname(input_file) or '.'
                success = run_suite(fh, handler_objects, host, port,
                                    prefix, force_ssl, failfast,
                                        test_directory=test_directory)
>>>>>>> REPLACE
```

**Answer 3:**
```python
### gabbi/runner.py
<<<<<<< SEARCH
def run_suite(handle, handler_objects, host, port, prefix, force_ssl=
    False,
             failfast=False):
    """Run the tests from the YAML in handle."""
    data = utils.load_yaml(handle)
    if force_ssl:
        if 'defaults' in data:
            data['defaults']['ssl'] = True
        else:
            data['defaults'] = {'ssl': True}

    loader = unittest.defaultTestLoader
    test_suite = suitemaker.test_suite_from_dict(
        loader, 'input', data, '.', host, port, None, None, prefix=
            prefix,
        handlers=handler_objects)

    result = ConciseTestRunner(
        verbosity=2, failfast=failfast).run(test_suite)
    return result.wasSuccessful()
=======
def run_suite(handle, handler_objects, host, port, prefix, force_ssl=
    False,
             failfast=False, test_directory='.'):
    """Run the tests from the YAML in handle."""
    data = utils.load_yaml(handle)
    if force_ssl:
        if 'defaults' in data:
            data['defaults']['ssl'] = True
        else:
            data['defaults'] = {'ssl': True}

    loader = unittest.defaultTestLoader
    test_suite = suitemaker.test_suite_from_dict(
        loader, 'input', data, test_directory, host, port, None, None,
        prefix=prefix, handlers=handler_objects)

    result = ConciseTestRunner(
        verbosity=2, failfast=failfast).run(test_suite)
    return result.wasSuccessful()
>>>>>>> REPLACE
```
```python
```

```
### gabbi/runner.py
<<<<<<< SEARCH
        for input_file in input_files:
            with open(input_file, 'r') as fh:
                success = run_suite(fh, handler_objects, host, port,
                                    prefix, force_ssl, failfast)
            if not failure:  # once failed, this is considered
                immutable
                failure = not success
            if failure and failfast:
                break
=======
        for input_file in input_files:
            test_directory = os.path.dirname(input_file)
            with open(input_file, 'r') as fh:
                success = run_suite(
                        fh, handler_objects, host, port, prefix, force_ssl,
                        failfast, test_directory=test_directory)
            if not failure:  # once failed, this is considered
                immutable
                failure = not success
            if failure and failfast:
                break
>>>>>>> REPLACE
```
```

**Answer 4:**
```python
### gabbi/runner.py
<<<<<<< SEARCH
def run_suite(handle, handler_objects, host, port, prefix, force_ssl=
    False,
             failfast=False):
=======
def run_suite(handle, handler_objects, host, port, prefix, force_ssl=
    False,
             failfast=False, test_directory='.'):
>>>>>>> REPLACE
```
```python
### gabbi/runner.py
<<<<<<< SEARCH
    test_suite = suitemaker.test_suite_from_dict(
        loader, 'input', data, '.', host, port, None, None, prefix=
            prefix,
        handlers=handler_objects)
=======
    test_suite = suitemaker.test_suite_from_dict(
        loader, 'input', data, test_directory, host, port, None, None,
        prefix=prefix, handlers=handler_objects)
>>>>>>> REPLACE
```
```python
### gabbi/runner.py
<<<<<<< SEARCH
    for input_file in input_files:
        with open(input_file, 'r') as fh:
            success = run_suite(fh, handler_objects, host, port,
                                prefix, force_ssl, failfast)
=======
    for input_file in input_files:
        # Determine test directory based on input file's location
        test_directory = os.path.dirname(os.path.abspath(input_file))
            or '.'
        with open(input_file, 'r') as fh:
```

```python
                success = run_suite(fh, handler_objects, host, port,
                                    prefix, force_ssl, failfast,
                                    test_directory=test_directory)
>>>>>>> REPLACE
```

**Answer 5:**
```python
### gabbi/runner.py
<<<<<<< SEARCH
import sys
import unittest
=======
import os
import sys
import unittest
>>>>>>> REPLACE
```
```python
### gabbi/runner.py
<<<<<<< SEARCH
def run_suite(handle, handler_objects, host, port, prefix, force_ssl=
    False,
              failfast=False):
=======
def run_suite(handle, handler_objects, host, port, prefix, force_ssl=
    False,
              failfast=False, test_directory='.'):
>>>>>>> REPLACE
```
```python
### gabbi/runner.py
<<<<<<< SEARCH
    test_suite = suitemaker.test_suite_from_dict(
        loader, 'input', data, '.', host, port, None, None, prefix=
            prefix,
        handlers=handler_objects)
=======
    test_suite = suitemaker.test_suite_from_dict(
        loader, 'input', data, test_directory, host, port, None, None,
        prefix=prefix, handlers=handler_objects)
>>>>>>> REPLACE
```
```python
### gabbi/runner.py
<<<<<<< SEARCH
        for input_file in input_files:
            with open(input_file, 'r') as fh:
                success = run_suite(fh, handler_objects, host, port,
                                    prefix, force_ssl, failfast)
=======
        for input_file in input_files:
            test_directory = os.path.dirname(input_file) or '.'
            with open(input_file, 'r') as fh:
                success = run_suite(fh, handler_objects, host, port,
                                    prefix, force_ssl, failfast,
                                    test_directory=test_directory)
>>>>>>> REPLACE
```

**Answer 6:**
```python
### gabbi/driver.py
<<<<<<< SEARCH
        file_suite = suitemaker.test_suite_from_dict(
```

```
            loader, test_base_name, suite_dict, path, host, port,
            fixture_module, intercept, prefix=prefix,
            test_loader_name=test_loader_name, handlers=handler_objects
                ,
            inner_fixtures=inner_fixtures)
=======
        test_file_dir = os.path.dirname(test_file)
        file_suite = suitemaker.test_suite_from_dict(
            loader, test_base_name, suite_dict, test_file_dir, host,
                port,
            fixture_module, intercept, prefix=prefix,
            test_loader_name=test_loader_name, handlers=handler_objects
                ,
            inner_fixtures=inner_fixtures)
>>>>>>> REPLACE
```
```python
### gabbi/runner.py
<<<<<<< SEARCH
def run_suite(handle, handler_objects, host, port, prefix, force_ssl=
    False,
              failfast=False):
=======
def run_suite(handle, handler_objects, host, port, prefix, force_ssl=
    False,
              failfast=False, test_directory='.'):
>>>>>>> REPLACE
```
```python
### gabbi/runner.py
<<<<<<< SEARCH
    test_suite = suitemaker.test_suite_from_dict(
        loader, 'input', data, '.', host, port, None, None, prefix=
            prefix,
        handlers=handler_objects)
=======
    test_suite = suitemaker.test_suite_from_dict(
        loader, 'input', data, test_directory, host, port, None, None,
        prefix=prefix, handlers=handler_objects)
>>>>>>> REPLACE
```
```python
### gabbi/runner.py
<<<<<<< SEARCH
                success = run_suite(fh, handler_objects, host, port,
                                    prefix, force_ssl, failfast)
=======
                test_directory = os.path.dirname(input_file)
                success = run_suite(fh, handler_objects, host, port,
                                    prefix, force_ssl, failfast,
                                       test_directory)
>>>>>>> REPLACE
```

**Answer 7:**
```python
### gabbi/runner.py
<<<<<<< SEARCH
import argparse
from importlib import import_module
import sys
import unittest
=======
import argparse
from importlib import import_module
```

```
import os
import sys
import unittest
>>>>>>> REPLACE
```
```python
### gabbi/runner.py
<<<<<<< SEARCH
def run_suite(handle, handler_objects, host, port, prefix, force_ssl=
    False,
              failfast=False):
    """Run the tests from the YAML in handle."""
    data = utils.load_yaml(handle)
    if force_ssl:
        if 'defaults' in data:
            data['defaults']['ssl'] = True
        else:
            data['defaults'] = {'ssl': True}

    loader = unittest.defaultTestLoader
    test_suite = suitemaker.test_suite_from_dict(
        loader, 'input', data, '.', host, port, None, None, prefix=
            prefix,
        handlers=handler_objects)
=======
def run_suite(handle, handler_objects, host, port, prefix, force_ssl=
    False,
              failfast=False, test_directory='.'):
    """Run the tests from the YAML in handle."""
    data = utils.load_yaml(handle)
    if force_ssl:
        if 'defaults' in data:
            data['defaults']['ssl'] = True
        else:
            data['defaults'] = {'ssl': True}

    loader = unittest.defaultTestLoader
    test_suite = suitemaker.test_suite_from_dict(
        loader, 'input', data, test_directory, host, port, None, None,
        prefix=prefix, handlers=handler_objects)
>>>>>>> REPLACE
```
```python
### gabbi/runner.py
<<<<<<< SEARCH
            with open(input_file, 'r') as fh:
                success = run_suite(fh, handler_objects, host, port,
                                    prefix, force_ssl, failfast)
            if not failure:  # once failed, this is considered
                immutable
                failure = not success
            if failure and failfast:
                break
=======
            test_dir = os.path.dirname(os.path.abspath(input_file))
            with open(input_file, 'r') as fh:
                success = run_suite(fh, handler_objects, host, port,
                                    prefix, force_ssl, failfast,
                                        test_directory=test_dir)
            if not failure:  # once failed, this is considered
                immutable
                failure = not success
            if failure and failfast:
                break
>>>>>>> REPLACE
```

```python
```
```

**Answer 8:**
```python
### gabbi/runner.py
<<<<<<< SEARCH
    data = utils.load_yaml(handle)
    if force_ssl:
        if 'defaults' in data:
            data['defaults']['ssl'] = True
        else:
            data['defaults'] = {'ssl': True}

    loader = unittest.defaultTestLoader
    test_suite = suitemaker.test_suite_from_dict(
        loader, 'input', data, '.', host, port, None, None, prefix=
            prefix,
        handlers=handler_objects)
=======
    data = utils.load_yaml(handle)
    if force_ssl:
        if 'defaults' in data:
            data['defaults']['ssl'] = True
        else:
            data['defaults'] = {'ssl': True}

    # Determine test directory based on input handle
    if hasattr(handle, 'name'):
        test_directory = os.path.dirname(os.path.abspath(handle.name))
    else:
        test_directory = '.'

    loader = unittest.defaultTestLoader
    test_suite = suitemaker.test_suite_from_dict(
        loader, 'input', data, test_directory, host, port, None, None,
        prefix=prefix, handlers=handler_objects)
>>>>>>> REPLACE
```

**Answer 9:**
```python
### gabbi/driver.py
<<<<<<< SEARCH
    for test_file in glob.iglob('%s/*.yaml' % path):
        if '_' in os.path.basename(test_file):
            warnings.warn(exception.GabbiSyntaxWarning(
                "'_' in test filename %s. This can break suite grouping
                    ."
                % test_file))
        if intercept:
            host = str(uuid.uuid4())
        suite_dict = utils.load_yaml(yaml_file=test_file)
        test_base_name = os.path.splitext(os.path.basename(test_file))
            [0]
        if all_test_base_name:
            test_base_name = '%s_%s' % (all_test_base_name,
                test_base_name)

        if require_ssl:
            if 'defaults' in suite_dict:
                suite_dict['defaults']['ssl'] = True
            else:
                suite_dict['defaults'] = {'ssl': True}

        file_suite = suitemaker.test_suite_from_dict(
```

```
                 loader, test_base_name, suite_dict, path, host, port,
                 fixture_module, intercept, prefix=prefix,
                 test_loader_name=test_loader_name, handlers=handler_objects
                     ,
                 inner_fixtures=inner_fixtures)
=======
    for test_file in glob.iglob('%s/*.yaml' % path):
        if '_' in os.path.basename(test_file):
            warnings.warn(exception.GabbiSyntaxWarning(
                "'_' in test filename %s. This can break suite grouping
                    ."
                % test_file))
        if intercept:
            host = str(uuid.uuid4())
        suite_dict = utils.load_yaml(yaml_file=test_file)
        test_base_name = os.path.splitext(os.path.basename(test_file))
            [0]
        if all_test_base_name:
            test_base_name = '%s_%s' % (all_test_base_name,
                test_base_name)

        if require_ssl:
            if 'defaults' in suite_dict:
                suite_dict['defaults']['ssl'] = True
            else:
                suite_dict['defaults'] = {'ssl': True}

        # Use absolute path of the YAML file's directory
        test_file_dir = os.path.dirname(os.path.abspath(test_file))
        file_suite = suitemaker.test_suite_from_dict(
            loader, test_base_name, suite_dict, test_file_dir, host,
                port,
            fixture_module, intercept, prefix=prefix,
            test_loader_name=test_loader_name, handlers=handler_objects
                ,
            inner_fixtures=inner_fixtures)
>>>>>>> REPLACE
```

**Answer 10:**
```python
### gabbi/runner.py
<<<<<<< SEARCH
import argparse
from importlib import import_module
import sys
import unittest
=======
import argparse
from importlib import import_module
import os
import sys
import unittest
>>>>>>> REPLACE
```
```python
### gabbi/runner.py
<<<<<<< SEARCH
def run_suite(handle, handler_objects, host, port, prefix, force_ssl=
    False,
              failfast=False):
=======
def run_suite(handle, handler_objects, host, port, prefix, force_ssl=
    False,
              failfast=False, test_directory='.'):
```

```
>>>>>>> REPLACE
```
```python
### gabbi/runner.py
<<<<<<< SEARCH
    test_suite = suitemaker.test_suite_from_dict(
        loader, 'input', data, '.', host, port, None, None, prefix=
            prefix,
        force_ssl=force_ssl, failfast=failfast,
        handlers=handler_objects)
=======
    test_suite = suitemaker.test_suite_from_dict(
        loader, 'input', data, test_directory, host, port, None, None,
        prefix=prefix, force_ssl=force_ssl, failfast=failfast,
        handlers=handler_objects)
>>>>>>> REPLACE
```
```python
### gabbi/runner.py
<<<<<<< SEARCH
                success = run_suite(fh, handler_objects, host, port,
                                    prefix, force_ssl, failfast)
=======
                test_dir = os.path.dirname(input_file)
                success = run_suite(fh, handler_objects, host, port,
                                    prefix, force_ssl, failfast,
                                    test_directory=test_dir)
>>>>>>> REPLACE
```
```python
### gabbi/case.py
<<<<<<< SEARCH
    path = os.path.join(self.test_directory, os.path.basename(filename)
        )
=======
    path = os.path.join(self.test_directory, filename)
>>>>>>> REPLACE
```

**Answer 11:**
```python
### gabbi/runner.py
<<<<<<< SEARCH
def run_suite(handle, handler_objects, host, port, prefix, force_ssl=
    False,
              failfast=False):
    """Run the tests from the YAML in handle."""
    data = utils.load_yaml(handle)
    if force_ssl:
        if 'defaults' in data:
            data['defaults']['ssl'] = True
        else:
            data['defaults'] = {'ssl': True}

    loader = unittest.defaultTestLoader
    test_suite = suitemaker.test_suite_from_dict(
        loader, 'input', data, '.', host, port, None, None, prefix=
            prefix,
        handlers=handler_objects)
=======
def run_suite(handle, handler_objects, host, port, prefix, force_ssl=
    False,
              failfast=False, test_directory='.'):
    """Run the tests from the YAML in handle."""
    data = utils.load_yaml(handle)
```

```python
    if force_ssl:
        if 'defaults' in data:
            data['defaults']['ssl'] = True
        else:
            data['defaults'] = {'ssl': True}

    loader = unittest.defaultTestLoader
    test_suite = suitemaker.test_suite_from_dict(
        loader, 'input', data, test_directory, host, port,
        None, None, prefix=prefix, handlers=handler_objects)
>>>>>>> REPLACE
```
```python
### gabbi/runner.py
<<<<<<< SEARCH
                success = run_suite(fh, handler_objects, host, port,
                                    prefix, force_ssl, failfast)
=======
                test_dir = os.path.dirname(input_file)
                success = run_suite(fh, handler_objects, host, port,
                                    prefix, force_ssl, failfast,
                                    test_directory=test_dir)
>>>>>>> REPLACE
```

**Answer 12:**
```python
### gabbi/runner.py
<<<<<<< SEARCH
import argparse
from importlib import import_module
import sys
import unittest
=======
import argparse
from importlib import import_module
import os
import sys
import unittest
>>>>>>> REPLACE
```
```python
### gabbi/runner.py
<<<<<<< SEARCH
    else:
        for input_file in input_files:
            with open(input_file, 'r') as fh:
                success = run_suite(fh, handler_objects, host, port,
                                    prefix, force_ssl, failfast)
=======
    else:
        for input_file in input_files:
            test_dir = os.path.dirname(os.path.abspath(input_file))
            with open(input_file, 'r') as fh:
                success = run_suite(fh, handler_objects, host, port,
                                    prefix, force_ssl, failfast,
                                    test_directory=test_dir)
>>>>>>> REPLACE
```
```python
### gabbi/runner.py
<<<<<<< SEARCH
def run_suite(handle, handler_objects, host, port, prefix, force_ssl=
    False,
              failfast=False):
```

```python
    """Run the tests from the YAML in handle."""
    data = utils.load_yaml(handle)
    if force_ssl:
        if 'defaults' in data:
            data['defaults']['ssl'] = True
        else:
            data['defaults'] = {'ssl': True}

    loader = unittest.defaultTestLoader
    test_suite = suitemaker.test_suite_from_dict(
        loader, 'input', data, '.', host, port, None, None, prefix=
            prefix,
        handlers=handler_objects)
=======
def run_suite(handle, handler_objects, host, port, prefix, force_ssl=
    False,
                failfast=False, test_directory='.'):
    """Run the tests from the YAML in handle."""
    data = utils.load_yaml(handle)
    if force_ssl:
        if 'defaults' in data:
            data['defaults']['ssl'] = True
        else:
            data['defaults'] = {'ssl': True}

    loader = unittest.defaultTestLoader
    test_suite = suitemaker.test_suite_from_dict(
        loader, 'input', data, test_directory, host, port, None, None,
        prefix=prefix, handlers=handler_objects)
>>>>>>> REPLACE
```

**Answer 13:**
```python
### gabbi/runner.py
<<<<<<< SEARCH
def run_suite(handle, handler_objects, host, port, prefix, force_ssl=
    False,
                failfast=False):
    """Run the tests from the YAML in handle."""
    data = utils.load_yaml(handle)
    if force_ssl:
        if 'defaults' in data:
            data['defaults']['ssl'] = True
        else:
            data['defaults'] = {'ssl': True}

    loader = unittest.defaultTestLoader
    test_suite = suitemaker.test_suite_from_dict(
        loader, 'input', data, '.', host, port, None, None, prefix=
            prefix,
        handlers=handler_objects)
=======
def run_suite(handle, handler_objects, host, port, prefix, force_ssl=
    False,
                failfast=False, test_directory='.'):
    """Run the tests from the YAML in handle."""
    data = utils.load_yaml(handle)
    if force_ssl:
        if 'defaults' in data:
            data['defaults']['ssl'] = True
        else:
            data['defaults'] = {'ssl': True}

    loader = unittest.defaultTestLoader
```

```python
        test_suite = suitemaker.test_suite_from_dict(
            loader, 'input', data, test_directory, host, port, None, None,
            prefix=prefix, handlers=handler_objects)
>>>>>>> REPLACE
```
```python
### gabbi/runner.py
<<<<<<< SEARCH
        for input_file in input_files:
            with open(input_file, 'r') as fh:
                success = run_suite(fh, handler_objects, host, port,
                                    prefix, force_ssl, failfast)
=======
        for input_file in input_files:
            test_dir = os.path.dirname(os.path.abspath(input_file))
            with open(input_file, 'r') as fh:
                success = run_suite(fh, handler_objects, host, port,
                                    prefix, force_ssl, failfast,
                                    test_directory=test_dir)
>>>>>>> REPLACE
```

**Answer 14:**
```python
### gabbi/runner.py
<<<<<<< SEARCH
        for input_file in input_files:
            with open(input_file, 'r') as fh:
                success = run_suite(fh, handler_objects, host, port,
                                    prefix, force_ssl, failfast)
=======
        for input_file in input_files:
            with open(input_file, 'r') as fh:
                success = run_suite(input_file, fh, handler_objects,
                    host, port,
                                    prefix, force_ssl, failfast)
>>>>>>> REPLACE
```
```python
### gabbi/runner.py
<<<<<<< SEARCH
def run_suite(handle, handler_objects, host, port, prefix, force_ssl=
    False,
              failfast=False):
    """Run the tests from the YAML in handle."""
    data = utils.load_yaml(handle)
    if force_ssl:
        if 'defaults' in data:
            data['defaults']['ssl'] = True
        else:
            data['defaults'] = {'ssl': True}

    loader = unittest.defaultTestLoader
    test_directory = '.'
    test_suite = suitemaker.test_suite_from_dict(
        loader, 'input', data, test_directory, host, port, None, None,
            prefix=prefix,
        handlers=handler_objects)
=======
def run_suite(input_file, handle, handler_objects, host, port, prefix,
    force_ssl=False,
              failfast=False):
    """Run the tests from the YAML in handle."""
    data = utils.load_yaml(handle)
    if force_ssl:
```

```
        if 'defaults' in data:
            data['defaults']['ssl'] = True
        else:
            data['defaults'] = {'ssl': True}

    loader = unittest.defaultTestLoader
    test_directory = os.path.dirname(input_file) if input_file else '.'
    test_suite = suitemaker.test_suite_from_dict(
        loader, 'input', data, test_directory, host, port, None, None,
            prefix=prefix,
        handlers=handler_objects)
>>>>>>> REPLACE
```
```python
### gabbi/runner.py
<<<<<<< SEARCH
    if not input_files:
        success = run_suite(sys.stdin, handler_objects, host, port,
                            prefix, force_ssl, failfast)
=======
    if not input_files:
        success = run_suite(None, sys.stdin, handler_objects, host,
            port,
                            prefix, force_ssl, failfast)
>>>>>>> REPLACE
```
```
--- END FILE ---

Please first summary and analyze the key differences between the
    trajectories, and then generate *SEARCH/REPLACE* edits to fix the
    issue.

Every *SEARCH/REPLACE* edit must use this format:
1. The file path
2. The start of search block: <<<<<<< SEARCH
3. A contiguous chunk of lines to search for in the existing source
    code
4. The dividing line: =======
5. The lines to replace into the source code
6. The end of the replace block: >>>>>>> REPLACE

Here is an example:
```
```python
### mathweb/flask/app.py
<<<<<<< SEARCH
from flask import Flask
=======
import math
from flask import Flask
>>>>>>> REPLACE
```
```
Please note that the *SEARCH/REPLACE* edit REQUIRES PROPER INDENTATION.
     If you would like to add the line '          print(x)', you must
    fully write that out, with all those spaces before the code!
Wrap the *SEARCH/REPLACE* edit in blocks ```python...```.
The summary of the key differences between the trajectories should be
    in the thinking part.
"
```

Listing 12: Example of emergent parallel scaling (`cdent__gabbi-191` in SWE-bench-extra), the prompt input part.

Listing 13 shows how the model successfully recognizes the correct answer candidate in Listing 12, and generates a novel patch with additional reasoning to solve the issue:

```
<think>
Okay, let's tackle this problem. The user is having an issue where when
    they specify a data file in their YAML test using the <@ syntax,
    the path isn't resolved relative to the YAML file's location.
    Instead, it's using the current working directory. They want the
    path to be relative to the YAML file so they can run tests from any
    directory.

First, I need to understand how the current code is handling the data
    file paths. The user mentioned that in _load_data_file, the code
    uses os.path.join(self.test_directory, os.path.basename(filename)).
    This takes the test_directory (which defaults to '.'), discards any
    directories in the filename, and appends the basename. That's why if
     the YAML is in /my/tests/test.yaml and the data is <@data.txt, it
    looks for ./data.txt instead of /my/tests/data.txt.

The key here is to make sure that the test_directory is set to the
    directory where the YAML file is located. Then, when joining with
    the filename (without basename), it would correctly resolve the
    relative path.

Looking at the provided code snippets, the main changes are in runner.
    py where the test_directory is determined. The user's proposed
    solutions involve modifying the run_suite function to accept a
    test_directory parameter, which is derived from the input file's
    directory.

In the original code, when running a suite from an input file, the
    test_directory is hardcoded to '.'. The various answers modify this
    by passing the directory of the input file as test_directory. For
    example, in Answer 10, they add the test_directory parameter to
    run_suite and set it to os.path.dirname(input_file). Then, in case.
    py, they change the path joining to use the filename directly
    instead of basename.

Wait, the critical part is in case.py's _load_data_file method. The
    original code uses os.path.basename(filename), which strips any
    directories from the filename. So if the YAML has <@test_data/file.
    txt, the basename is file.txt, and it's joined with test_directory.
    But if test_directory is the YAML's directory, then it should be
    YAML_DIR/test_data/file.txt. However, the current code would do
    test_directory + file.txt, which would be YAML_DIR/file.txt. That's
    incorrect. So the problem is that the code is taking the basename,
    thus ignoring any subdirectories in the filename.

Ah, so the fix here is to not take the basename. Instead, the filename
    after <@ should be treated as a relative path from the YAML's
    directory. So in case.py's _load_data_file, instead of using os.path
    .basename(filename), we should use the filename as-is. Then, os.path
    .join(self.test_directory, filename) would correctly resolve the
    path relative to the YAML's directory.

But wait, the filename could be an absolute path. For example, if the
    YAML has <@/absolute/path/file.txt, then os.path.join would just use
     the absolute path. However, the documentation says that the
    filename is relative to the YAML file. So absolute paths might not
    be intended. But the code should handle that.

So combining two changes: in runner.py, set test_directory to the YAML
    file's directory, and in case.py, stop using basename and use the
    full filename.
```

```
Looking at Answer 10, there's a change in case.py from os.path.basename
    (filename) to filename. That's exactly the fix needed. Then, in
    runner.py, when running the suite, pass the test_directory as the
    directory of the input file.

So the key files to modify are:

1. gabbi/runner.py: When calling run_suite, pass test_directory as the
    directory of the input file. This involves modifying the run_suite
    function to accept test_directory as a parameter and then passing it
     when the input file is processed.

2. gabbi/case.py: Change the path construction from using os.path.
    basename(filename) to using the filename directly. This allows
    subdirectories in the filename to be included in the path.

Looking through the provided code snippets, the one that does both is
    Answer 10. Let me check:

In Answer 10, the runner.py changes add the test_directory parameter
    and set it to os.path.dirname(input_file). Then, in case.py, the
    path is changed to os.path.join(self.test_directory, filename). That
     would correctly resolve the path relative to the YAML's directory,
    including any subdirectories in the filename.

But wait, the original code in case.py uses os.path.basename(filename).
     So changing that to filename means that if the input is 'test_data/
    file.txt', it's joined as test_directory/test_data/file.txt. If
    test_directory is the YAML's directory, that's correct.

...  [intermediate reasoning omitted for display brevity] ...

But perhaps better to use os.path.abspath and then dirname to handle
    relative paths correctly. But maybe the existing code already works.
     For example, if the input_file is 'tests/test.yaml', then os.path.
    dirname would return 'tests', which when joined with os.getcwd()
    would be correct. Wait, no. os.path.dirname just gives the directory
     part of the path. So if input_file is a relative path, like 'tests/
    test.yaml', then test_directory would be 'tests', and when os.path.
    join is called, it's relative to the current working directory. But
    the code in suitemaker.test_suite_from_dict might expect an absolute
     path. However, the original code uses '.' as the test_directory,
    which is relative. So perhaps using the directory of the input file
    as test_directory is sufficient.

Another thing to consider: when the YAML is loaded from stdin (like via
     redirect), the test_directory remains '.'. That's acceptable, as
    the documentation mentions that when using <, the filename isn't
    seen by Gabbi, so it's not possible to resolve the path relative to
    the YAML's original location. So in that case, the test_directory
    defaults to '.'.

Now, in case.py:

Original line:

path = os.path.join(self.test_directory, os.path.basename(filename))

Modified line:

path = os.path.join(self.test_directory, filename)

So the SEARCH/REPLACE block for case.py:

### gabbi/case.py
```

```
<<<<<<< SEARCH
    path = os.path.join(self.test_directory, os.path.basename(filename)
        )
=======
    path = os.path.join(self.test_directory, filename)
>>>>>>> REPLACE

Putting all these together, the final edits are:

1. Add import os in runner.py.

2. Modify run_suite to accept test_directory and pass it to
   test_suite_from_dict.

3. In the loop over input_files, compute test_directory as the
   directory of input_file and pass it to run_suite.

4. Modify case.py's path construction.

So the final answer would include these four SEARCH/REPLACE blocks.
◁/think▷

'''python
### gabbi/runner.py
<<<<<<< SEARCH
import argparse
from importlib import import_module
import sys
import unittest
=======
import argparse
from importlib import import_module
import os
import sys
import unittest
>>>>>>> REPLACE
'''

'''python
### gabbi/runner.py
<<<<<<< SEARCH
def run_suite(handle, handler_objects, host, port, prefix, force_ssl=
    False,
            failfast=False):
=======
def run_suite(handle, handler_objects, host, port, prefix, force_ssl=
    False,
            failfast=False, test_directory='.'):
>>>>>>> REPLACE
'''

'''python
### gabbi/runner.py
<<<<<<< SEARCH
    test_suite = suitemaker.test_suite_from_dict(
        loader, 'input', data, '.', host, port, None, None, prefix=
            prefix,
        handlers=handler_objects)
=======
    test_suite = suitemaker.test_suite_from_dict(
        loader, 'input', data, test_directory, host, port, None, None,
        prefix=prefix, handlers=handler_objects)
>>>>>>> REPLACE
'''
```

```
```python
### gabbi/runner.py
<<<<<<< SEARCH
    else:
        for input_file in input_files:
            with open(input_file, 'r') as fh:
                success = run_suite(fh, handler_objects, host, port,
                                    prefix, force_ssl, failfast)
=======
    else:
        for input_file in input_files:
            test_directory = os.path.dirname(input_file) or '.'
            with open(input_file, 'r') as fh:
                success = run_suite(fh, handler_objects, host, port,
                                    prefix, force_ssl, failfast,
                                    test_directory=test_directory)
>>>>>>> REPLACE
```

```python
### gabbi/case.py
<<<<<<< SEARCH
    path = os.path.join(self.test_directory, os.path.basename(filename)
        )
=======
    path = os.path.join(self.test_directory, filename)
>>>>>>> REPLACE
```

These changes:
1. Add 'os' import needed for path handling
2. Modify run_suite to accept a test_directory parameter
3. Use the input file's directory as the test_directory when processing
   files
4. Update data file path resolution to preserve directory structure
   from YAML relative paths
5. Remove basename handling that was stripping directory information
   from data file paths

The net effect is that data files specified with '<@filename' in YAML
   tests will now be resolved relative to their YAML file's directory,
   supporting both flat and nested directory structures for test data.
   When tests are run from stdin (using '<'), the default directory
   remains the current working directory as before.
```

Listing 13: Example of emergent parallel scaling (`cdent‿gabbi-191` in SWE-bench-extra), the model response part.

## G  ANALYSIS FOR SWE-AGENTS

### G.1  STAGE ANNOTATION FOR SWE-AGENT TRAJECTORIES

In this section, we present how we use a frontier LLM to annotate the SWE-Agent stage to which each interaction turn within the trajectory rollout belongs. While we have briefly introduced the five stages suggested in the prompt of the SWE-Agent prompt in Section 4, we attach the excerpt in Listing 14 for greater clarity:

```
...
Follow these steps to resolve the issue:
1. As a first step, it might be a good idea to find and read code
   relevant to the <pr_description>
2. Create a script to reproduce the error and execute it with 'python <
   filename.py>' using the bash tool, to confirm the error
```

```
3. Edit the source code of the repo to resolve the issue
4. Rerun your reproduce script and confirm that the error is fixed!
5. Think about edgecases and make sure your fix handles them as well
...
```

Listing 14: The excerpt of the five-stage declaration in the SWE-Agent prompt.

It should be noted that the agent could flexibly transit across the five stages during its working process. For example, after Stage 4 when the agent rerun the test script, possibilities are that erroneous information remains, and this is when the agent goes back to Stage 3 to refine its code repair with reflection; Similar backtracing behavior could be observed from Stage 5 to Stage 3 as well, where the initial code repair has proven correct under the initial test script the agent composes in Stage 2, but fails some edge testcase the agent proposes in Stage 5.

To further analyze the BugFixer and the reflection skill prior, we need to realize which stage each turn along the SWE-Agent trajectory belongs to. As no strict boundaries or special prompt notes are set between each consecutive stage, we leverage an LLM for annotation. The annotation system prompt we set in kimi-k2-0711-preview is shown in Listing 15:

```
You are a professional inspector that can analyze the provided agentic
    interaction trajectory.

The trajectory you are going to analyze is made by an agent that
    interacts with a computer to solve tasks. This agent has access to
    the following functions:

---- BEGIN FUNCTION #1: bash ----
Description: Execute a bash command in the terminal.

Parameters:
  (1) command (string, required): The bash command to execute. Can be
      empty to view additional logs when previous exit code is `-1`. Can
       be `ctrl+c` to interrupt the currently running process.
---- END FUNCTION #1 ----

---- BEGIN FUNCTION #2: submit ----
Description: Finish the interaction when the task is complete OR if the
     assistant cannot proceed further with the task.
No parameters are required for this function.
---- END FUNCTION #2 ----

---- BEGIN FUNCTION #3: str_replace_editor ----
Description: Custom editing tool for viewing, creating and editing
    files
* State is persistent across command calls and discussions with the
    user
* If `path` is a file, `view` displays the result of applying `cat -n`.
     If `path` is a directory, `view` lists non-hidden files and
    directories up to 2 levels deep
* The `create` command cannot be used if the specified `path` already
    exists as a file
* If a `command` generates a long output, it will be truncated and
    marked with `<response clipped>`
* The `undo_edit` command will revert the last edit made to the file at
     `path`

Notes for using the `str_replace` command:
* The `old_str` parameter should match EXACTLY one or more consecutive
    lines from the original file. Be mindful of whitespaces!
* If the `old_str` parameter is not unique in the file, the replacement
     will not be performed. Make sure to include enough context in `
    old_str` to make it unique
* The `new_str` parameter should contain the edited lines that should
    replace the `old_str`
```

```
Parameters:
  (1) command (string, required): The commands to run. Allowed options
      are: 'view', 'create', 'str_replace', 'insert', 'undo_edit'.
Allowed values: ['view', 'create', 'str_replace', 'insert', 'undo_edit
  ']
  (2) path (string, required): Absolute path to file or directory, e.g.
      '/repo/file.py' or '/repo'.
  (3) file_text (string, optional): Required parameter of 'create'
      command, with the content of the file to be created.
  (4) old_str (string, optional): Required parameter of 'str_replace'
      command containing the string in 'path' to replace.
  (5) new_str (string, optional): Optional parameter of 'str_replace'
      command containing the new string (if not given, no string will be
       added). Required parameter of 'insert' command containing the
      string to insert.
  (6) insert_line (integer, optional): Required parameter of 'insert'
      command. The 'new_str' will be inserted AFTER the line '
      insert_line' of 'path'.
  (7) view_range (array, optional): Optional parameter of 'view'
      command when 'path' points to a file. If none is given, the full
      file is shown. If provided, the file will be shown in the
      indicated line number range, e.g. [11, 12] will show lines 11 and
      12. Indexing at 1 to start. Setting '[start_line, -1]' shows all
      lines from 'start_line' to the end of the file.
---- END FUNCTION #3 ----

The agent was instructed with the following:

* A python code repository has been uploaded in the directory /testbed.
* Implement the necessary changes to the repository so that the
    requirements specified in the <pr_description> are met.
* All changes to any of the test files described in the <pr_description
    > have already been taken care of. This means no need to modify the
    testing logic or any of the tests in any way.
* Make the minimal changes to non-tests files in the /testbed directory
     to ensure the <pr_description> is satisfied.

The agent was suggested to follow the following steps to resolve the
    issue:
1. As a first step, it might be a good idea to find and read code
    relevant to the <pr_description>
2. Create a script to reproduce the error and execute it with 'python <
    filename.py>' using the bash tool, to confirm the error
3. Edit the source code of the repo to resolve the issue
4. Rerun your reproduce script and confirm that the error is fixed!
5. Think about edgecases and make sure your fix handles them as well
The agent was encouraged to think thoroughly, and it's fine if it's
    very long.

You are going to inspect this agent's interaction trajectory with a
    computer to solve the given task in the <pr_description>. One turn
    of interaction contains a pair of OBSERVATION and ACTION, where the
    OBSERVATION comes from the computer, and the ACTION is taken by the
    agent.

For each turn of interaction, determine which step (of the
    aforementioned five) this turn belongs to. Output a single number
    (1~5) ONLY in a separate line as your classification (DO NOT OUTPUT
    ANY OTHER WORDS THAN THE DIGIT).

You can think before make the inspection. When thinking, wrap your
    thought with <think> and </think>. Don't forget to output your final
     inspection after thinking.
```

Listing 15: The annotation prompt for SWE-Agent stages.

To provide a clearer understanding of the trajectory, we incorporate most of the tool descriptions and instructions from the SWE-Agent system prompt into the annotation system prompt. The annotation is conducted in a multi-round manner, leveraging the agent's previous actions and observations, as well as the stage classifications of earlier turns, to better exploit contextual information. At the $i$-th round of annotation, the observation–action pair from turn $i$ of the SWE-Agent trajectory is appended as input, and the annotator is expected to output the corresponding stage classification.

## G.2 COMPARATIVE STUDY

Based on the automatic stage annotation in the above section, we present a comparative study by inspecting the performance on `sympy__sympy-20590` among the Kimi-Dev under Agentless, and each of the Base, MT, SFT, and RL priors with SWE-Agent adaptation.

The problem statement of `sympy__sympy-20590` is listed in Listing 16:

```
Symbol instances have __dict__ since 1.7?
In version 1.6.2 Symbol instances had no `__dict__` attribute
```python
>>> sympy.Symbol('s').__dict__
---------------------------------------------------------------------

AttributeError Traceback (most recent call last)
<ipython-input-3-e2060d5eec73> in <module>
----> 1 sympy.Symbol('s').__dict__

AttributeError: 'Symbol' object has no attribute '__dict__'
>>> sympy.Symbol('s').__slots__
('name',)
```

This changes in 1.7 where `sympy.Symbol('s').__dict__` now exists (and
    returns an empty dict)
I may misinterpret this, but given the purpose of `__slots__`, I assume
    this is a bug, introduced because some parent class accidentally
    stopped defining `__slots__`.
```

Listing 16: The problem statement of `sympy__sympy-20590`.

It is observed that the main difficulty in resolving the issue lies in the realization of the "*some* parent class" referenced in the problem. In fact, the hints text of this problem, which reflects the discussion of the developers under the original issue, reveals a much more in-depth investigation into the issue (Listing 17):

```
It seems that Basic now inherits `DefaultPrinting` which I guess doesn'
    t have slots. I'm not sure if it's a good idea to add `__slots__` to
     that class as it would then affect all subclasses.

...

Using slots can break multiple inheritance but only if the slots are
    non-empty I guess. Maybe this means that any mixin should always
    declare empty slots or it won't work properly with subclasses that
    have slots...

I see that `EvalfMixin` has `__slots__ = ()`.
I guess we should add empty slots to DefaultPrinting then.
```

Listing 17: The excerpted hints text of `sympy__sympy-20590`.

According to the discussion, it is clear that the code repair would be to "add empty slots to Default-Printing", which naturally leads to the navigation towards the file related to the implementation of the printer (`sympy/core/_print_helpers.py`, which is also the file updated by the ground-truth patch.) However, the `hints_text` information in the test set is *not* allowed to be used in the problem-solving process, which challenges the reasoner or the agent to figure out "the parent class that stopped defining `__slots__`" autonomously.

We first examine Kimi-Dev under Agentless. None of the 40 runs succeeded in producing the correct file localization. In most cases, the updates are made to `sympy/core/symbol.py`, which is a plausible choice since the reported problem is triggered by `sympy.Symbol('s')`, and `symbol.py` should contain the definition of the `Symbol` class. This line of reasoning, however, diverges from the correct solution. A small number of runs attempt alternative localizations, such as `sympy/core/{basic.py, expr.py}`, but these are also incorrect.

The limitation of the Agentless framework becomes more obvious when the focus is shifted to the inspection over the SWE-Agent trajectories. For all of the Base, MT, SFT, and RL priors with SWE-Agent adaptation, the resulting trajectories always contain a test-reproducing phase (marked as Stage "2" by the annotator), with the generated script reads like:

```
from sympy import Symbol

# Try to access __dict__ on a Symbol instance
s = Symbol('s')
print("Symbol.__slots__:", s.__slots__)
try:
    print("Symbol.__dict__:", s.__dict__)
except AttributeError as e:
    print("Error:", e)
```

Listing 18: An exampled reproducing script in Stage 2 within the SWE-Agent trajectories in `sympy__sympy-20590`.

It is trivial for an agent to generate a reproducing script of this form, as it directly transcribes the case described in the problem statement. Executing the script provides an unambiguous signal of whether the issue is resolved: the attribute `s.__dict__` should no longer exist, and the expected `Error:` message should appear in the standard error output. Indeed, observing the initial completion of Stage 3 across all trajectories reveals that none of the agents succeeded on their first attempt. However, the incomplete status is consistently exposed in Stage 4, when the script is rerun and `s.__dict__` still persists. This progression highlights the advantage of the SWE-Agent framework over the Agentless paradigm, as the agent is afforded additional opportunities for iterative refinement rather than being limited to a single attempt.

We continue to analyze the behavior of the agents adapted from the Base, MT, and SFT priors. Due to the difficulty of the issue, it is non-trivial to correctly localize the error through their agentic reasoning: the interactions show that the agents were switching among several file candidates, but failed to reach the correct one. The only exception comes from the agent adapted from the RL prior, which marks its unique success on this instance through accurate agentic reasoning. We attach the intermediate process from the initial Stage-3 moment to the Stage-3 revisit (refining the initial code repair) within the successful trajectory, tagged by the stage annotator:

```
...

OBSERVATION:
Symbol.__slots__: ('name',)
Symbol.__dict__: {}

ACTION:
Hmm, it seems like the issue is still there. Let's check if there are
    any other classes in the hierarchy that might be missing '__slots__
    ':

<function=bash>
```

```
<parameter=command>python -c "from sympy import Symbol; s = Symbol('s')
    ; for cls in type(s).__mro__: print(cls.__name__, 'has __slots__:',
    hasattr(cls, '__slots__'))"</parameter>
</function>
```

**OBSERVATION:**
```
File "<string>", line 1
    from sympy import Symbol; s = Symbol('s'); for cls in type(s).
        __mro__: print(cls.__name__, 'has __slots__:', hasattr(cls, '
        __slots__'))
                                                     ^
SyntaxError: invalid syntax
```

**ACTION:**
Let's try a different approach:

```
<function=str_replace_editor>
<parameter=command>create</parameter>
<parameter=path>/testbed/check_slots.py</parameter>
<parameter=file_text>
from sympy import Symbol

s = Symbol('s')
for cls in type(s).__mro__:
    print(f"{cls.__name__:15} has __slots__: {hasattr(cls, '__slots__')
        }")
    if hasattr(cls, '__slots__'):
        print(f"{' '*16}__slots__ value: {cls.__slots__}")
</parameter>
</function>
```

**OBSERVATION:**
```
File created successfully at: /testbed/check_slots.py
```

**ACTION:**
```
<function=bash>
<parameter=command>python /testbed/check_slots.py</parameter>
</function>
```

**OBSERVATION:**
```
Symbol          has __slots__: True
                __slots__ value: ('name',)
AtomicExpr      has __slots__: True
                __slots__ value: ()
Atom            has __slots__: True
                __slots__ value: ()
Expr            has __slots__: True
                __slots__ value: ()
Boolean         has __slots__: True
                __slots__ value: ()
Basic           has __slots__: True
                __slots__ value: ('_mhash', '_args', '_assumptions')
Printable       has __slots__: False
EvalfMixin      has __slots__: True
                __slots__ value: ()
object          has __slots__: False
```

**ACTION:**
Now I see the issue! `Printable` doesn't have `__slots__` defined. Let'
    s check the `Printable` class:

```
<function=str_replace_editor>
<parameter=command>view</parameter>
<parameter=path>/testbed/sympy/core/_print_helpers.py</parameter>
<parameter=view_range>[1, 30]</parameter>
```

```
</function>
```

**OBSERVATION:**
Here's the result of running `cat -n` on /testbed/sympy/core/
    _print_helpers.py:
```
 1  """
 2  Base class to provide str and repr hooks that `init_printing`
     can overwrite.
 3
 4  This is exposed publicly in the `printing.defaults` module,
 5  but cannot be defined there without causing circular imports.
 6  """
 7
 8  class Printable:
 9      """
10      The default implementation of printing for SymPy classes.
11
12      This implements a hack that allows us to print elements of
    built-in
13      Python containers in a readable way. Natively Python uses
    ``repr()``
14      even if ``str()`` was explicitly requested. Mix in this
    trait into
15      a class to get proper default printing.
16
17      This also adds support for LaTeX printing in jupyter
    notebooks.
18      """
19
20      # Note, we always use the default ordering (lex) in __str__
     and __repr__,
21      # regardless of the global setting. See issue 5487.
22      def __str__(self):
23          from sympy.printing.str import sstr
24          return sstr(self, order=None)
25
26      __repr__ = __str__
27
28      def _repr_disabled(self):
29          """
30          No-op repr function used to disable jupyter display
    hooks.
```

**ACTION:**
We need to add `__slots__ = ()` to the `Printable` class as well:

```
<function=str_replace_editor>
<parameter=command>str_replace</parameter>
<parameter=path>/testbed/sympy/core/_print_helpers.py</parameter>
<parameter=old_str>
class Printable:
    """
    The default implementation of printing for SymPy classes.

    This implements a hack that allows us to print elements of built-in
    Python containers in a readable way. Natively Python uses ``repr()
        ``
    even if ``str()`` was explicitly requested. Mix in this trait into
    a class to get proper default printing.

    This also adds support for LaTeX printing in jupyter notebooks.
    """
</parameter>
<parameter=new_str>
class Printable:
```

```
    """
    The default implementation of printing for SymPy classes.

    This implements a hack that allows us to print elements of built-in
    Python containers in a readable way. Natively Python uses ``repr()
        ``
    even if ``str()`` was explicitly requested. Mix in this trait into
    a class to get proper default printing.

    This also adds support for LaTeX printing in jupyter notebooks.
    """

    __slots__ = ()
</parameter>
</function>

...
```

Listing 19: The intermediate reflection process between the two Stage-3 moment of the successful trajectory in `sympy__sympy-20590`.

Listing 19 shows the advanced reflection capability through slot checking along the entire class hierarchy, which successfully leads to the observation of the erroneous `Printable`. The following behavior from the agent includes the routine of test rerun, and the observation of passing the test script (Listing 18). This chain of actions recovers and resonates with the developer discussions in `hints_text` (Listing 17), showing the superior reflection skill in the RL prior over other priors.

### G.3 PATTERN COMPARISON BETWEEN SFT WITH EXPERT DATA AND END-TO-END RL

In this section, we compare the patterns of the interaction trajectories of the RL prior that undergoes (i) $1.5 \times 2^{28}$ SWE-Agent expert SFT tokens (later referred as "heavy SFT") and (ii) $2^{21}$ SWE-Agent expert SFT tokens + end-to-end SWE-Agent RL (later referred as "light SFT + end-to-end RL") as a qualitative study for Figure 7.

We select an instance solved by both methods, `matplotlib__matplotlib-13989`, as case study. We attach the actions along the two interaction trajectories for comparison.

The action patterns of the heavy SFT model read as Listing 20:

```
ACTION 0:
I'll help you implement the necessary changes to fix the issue where `
    hist()` doesn't respect `range=...` when `density=True`. Let's
    follow the steps:

1. First, let's find and read the relevant code:

<function=bash>
<parameter=command>find /testbed -type f -name "*.py" | grep -v "test_
    "</parameter>
</function>

ACTION 1:
Let's look for the histogram implementation since that's what we need
    to fix. The PR mentions that the bug is in the `hist()` function
    when `density=True` and `range=...` is specified.

<function=bash>
<parameter=command>grep -r "def hist" --include="*.py" /testbed | grep
    -v "test_"</parameter>
</function>

ACTION 2:
The main histogram implementation appears to be in `/testbed/lib/
    matplotlib/axes/_axes.py`. Let's look at that file:
```

```
<function=str_replace_editor>
<parameter=command>view</parameter>
<parameter=path>/testbed/lib/matplotlib/axes/_axes.py</parameter>
</function>
```

**ACTION 3:**
Let me look at the histogram implementation specifically:

```
<function=str_replace_editor>
<parameter=command>view</parameter>
<parameter=path>/testbed/lib/matplotlib/axes/_axes.py</parameter>
<parameter=view_range>[6365, 6899]</parameter>
</function>
```

**ACTION 4:**
Let me also look at the `histogram_bin_edges` function since it's used in the code:

```
<function=str_replace_editor>
<parameter=command>view</parameter>
<parameter=path>/testbed/lib/matplotlib/axes/_axes.py</parameter>
<parameter=view_range>[40, 60]</parameter>
</function>
```

**ACTION 5:**
2. Let's create a script to reproduce the error:

```
<function=str_replace_editor>
<parameter=command>create</parameter>
<parameter=path>/testbed/reproduce.py</parameter>
...
```

**ACTION 6:**
Let's run it to confirm the issue:

```
<function=bash>
<parameter=command>python /testbed/reproduce.py</parameter>
</function>
```

**ACTION 7:**
The issue is confirmed: when `density=True` is specified, the `range=(0, 1)` parameter is not respected. The bins should start at 0 and end at 1, but they don't.

3. Let's fix the issue. Looking at the code, the problem is in the `hist` method and specifically around line 6689-6690:

```python
if density and not stacked:
    hist_kwargs = dict(density=density)
```

The issue is that when `density=True`, the `range` parameter gets lost because we're not including it in the `hist_kwargs`. Let's modify the code to fix this:

```
<function=str_replace_editor>
<parameter=command>str_replace</parameter>
<parameter=path>/testbed/lib/matplotlib/axes/_axes.py</parameter>
...
```

**ACTION 8:**
4. Let's run the reproduce script again to confirm the fix:

```
<function=bash>
<parameter=command>python /testbed/reproduce.py</parameter>
</function>

ACTION 9:
Great! The fix works. Now the bins start at 0 and end at 1 as expected.

5. Let's think about edge cases:

1. When 'range=None' - This should use the min/max of the data, which
   is already handled correctly by the existing code
2. When 'range' is specified but 'density=False' - The range should be
    respected, which it already is
3. When 'range' is specified and 'density=True' - The range should be
    respected, which our fix now ensures
4. When 'bins' is a sequence - The range should be ignored, which is
    already handled by the code
5. When 'range' includes values outside the data range - The histogram
    should include empty bins for the range outside the data, which is
    already handled by numpy's histogram function

Let me test some of these edge cases to be sure:

<function=str_replace_editor>
<parameter=command>str_replace</parameter>
<parameter=path>/testbed/reproduce.py</parameter>
...

ACTION 10:
<function=bash>
<parameter=command>python /testbed/reproduce.py</parameter>
</function>

ACTION 11:
All test cases look good:

1. When 'range=(0, 1)' and 'density=True', the bins start at 0 and end
    at 1
2. When 'range=None' and 'density=True', the bins use the min/max of
   the data
3. When 'range=(0, 1)' and 'density=False', the bins start at 0 and end
    at 1
4. When 'bins' is a sequence, the range parameter is ignored and the
   sequence is used directly
5. When 'range=(-0.5, 1.5)', the bins start at -0.5 and end at 1.5

The fix looks good and handles all edge cases correctly. Let me submit
    the changes:

<function=submit>
</function>
```

Listing 20: The interaction made by the RL prior with $1.5 \times 2^{28}$ SWE-Agent SFT tokens on `matplotlib_matplotlib-13989`.

The action patterns of the light SFT + end-to-end RL model read as Listing 21:

```
ACTION 0:
Understood. Let's tackle this problem step by step.

First, I need to find the relevant code in the /testbed directory. The
    user mentioned that the code is in /testbed, so I'll start by
    listing the files there.

<function=str_replace_editor>
```

```
<parameter=command>view</parameter>
<parameter=path>/testbed</parameter>
</function>

ACTION 1:
Hmm, the /testbed directory has a lot of files, but I need to focus on
    the code related to the hist function. The hist function is part of
    matplotlib.pyplot, so I should look for the pyplot module in the /
    testbed/lib/matplotlib directory.

<function=str_replace_editor>
<parameter=command>view</parameter>
<parameter=path>/testbed/lib/matplotlib/pyplot.py</parameter>
</function>

ACTION 2:
Okay, I'll look at the hist function in pyplot.py.

<function=str_replace_editor>
<parameter=command>view</parameter>
<parameter=path>/testbed/lib/matplotlib/pyplot.py</parameter>
<parameter=view_range>[2610, 2625]</parameter>
</function>

ACTION 3:
The hist function in pyplot.py simply calls the hist method of the
    current axes. So the actual implementation is in the Axes class.

I'll look for the Axes.hist method in the /testbed/lib/matplotlib/axes
    directory.

<function=str_replace_editor>
<parameter=command>view</parameter>
<parameter=path>/testbed/lib/matplotlib/axes</parameter>
</function>

ACTION 4:
The Axes.hist method is likely in _axes.py, which is where most of the
    plotting methods are defined.

<function=str_replace_editor>
<parameter=command>view</parameter>
<parameter=path>/testbed/lib/matplotlib/axes/_axes.py</parameter>
</function>

ACTION 5:
I'll look at the hist method in _axes.py.

<function=str_replace_editor>
<parameter=command>view</parameter>
<parameter=path>/testbed/lib/matplotlib/axes/_axes.py</parameter>
<parameter=view_range>[6365, 6400]</parameter>
</function>

ACTION 6:
I'll look at more of the hist method to see where the range is handled.

<function=str_replace_editor>
<parameter=command>view</parameter>
<parameter=path>/testbed/lib/matplotlib/axes/_axes.py</parameter>
<parameter=view_range>[6400, 6450]</parameter>
</function>

ACTION 7:
I'll look at the code where the bins are calculated.
```

```
<function=str_replace_editor>
<parameter=command>view</parameter>
<parameter=path>/testbed/lib/matplotlib/axes/_axes.py</parameter>
<parameter=view_range>[6450, 6500]</parameter>
</function>
```

**ACTION 8:**
I'll look at the code where the bins are calculated.

```
<function=str_replace_editor>
<parameter=command>view</parameter>
<parameter=path>/testbed/lib/matplotlib/axes/_axes.py</parameter>
<parameter=view_range>[6500, 6550]</parameter>
</function>
```

**ACTION 9:**
I'll look at the code where the bins are calculated.

```
<function=str_replace_editor>
<parameter=command>view</parameter>
<parameter=path>/testbed/lib/matplotlib/axes/_axes.py</parameter>
<parameter=view_range>[6550, 6600]</parameter>
</function>
```

**ACTION 10:**
I'll look at the code where the bins are calculated.

```
<function=str_replace_editor>
<parameter=command>view</parameter>
<parameter=path>/testbed/lib/matplotlib/axes/_axes.py</parameter>
<parameter=view_range>[6600, 6650]</parameter>
</function>
```

**ACTION 11:**
I'll look at the code where the bins are calculated.

```
<function=str_replace_editor>
<parameter=command>view</parameter>
<parameter=path>/testbed/lib/matplotlib/axes/_axes.py</parameter>
<parameter=view_range>[6650, 6700]</parameter>
</function>
```

**ACTION 12:**
Ah, here's the key part. The code calculates the bins using np.
    histogram, and if the range is specified, it's passed to
    histogram_bin_edges. But when density is True, the code may be
    overriding the range.

Wait, in the code:

```
if not input_empty and len(x) > 1:
    if weights is not None:
        _w = np.concatenate(w)
    else:
        _w = None

    bins = histogram_bin_edges(np.concatenate(x),
                                bins, bin_range, _w)
else:
    hist_kwargs['range'] = bin_range
```

But when density is True, the code may be using the default range of
    the data, not the specified range.

```
Wait, no. The code for calculating the bins is:

if bins is None:
    bins = rcParams['hist.bins']

if bin_range is not None:
    bin_range = self.convert_xunits(bin_range)

if bin_range is None:
    xmin = np.inf
    xmax = -np.inf
    for xi in x:
        if len(xi):
            xmin = min(xmin, np.nanmin(xi))
            xmax = max(xmax, np.nanmax(xi))
    if not np.isnan([xmin, xmax]).any() and not (xmin > xmax):
        bin_range = (xmin, xmax)

# If bins are not specified either explicitly or via range,
# we need to figure out the range required for all datasets,
# and supply that to np.histogram.
if not input_empty and len(x) > 1:
    if weights is not None:
        _w = np.concatenate(w)
    else:
        _w = None

    bins = histogram_bin_edges(np.concatenate(x),
                               bins, bin_range, _w)
else:
    hist_kwargs['range'] = bin_range

density = bool(density) or bool(normed)
if density and not stacked:
    hist_kwargs = dict(density=density)

# List to store all the top coordinates of the histograms
tops = []
mlast = None
# Loop through datasets
for i in range(nx):
    # this will automatically overwrite bins,
    # so that each histogram uses the same bins
    m, bins = np.histogram(x[i], bins, weights=w[i], **hist_kwargs)
    m = m.astype(float)  # causes problems later if it's an int
    if mlast is None:

But when density is True, the code passes density=True to np.histogram.
    However, when density is True, np.histogram normalizes the counts,
   but the bin edges are still determined by the range.

Wait, but the problem is that when density=True, the range is not
   respected. The user's code is:

_, bins, _ = plt.hist(np.random.rand(10), "auto", range=(0, 1), density
   =True)
print(bins)

and the output is:

[0.00331535 0.18930174 0.37528813 0.56127453 0.74726092 0.93324731]

which is not from 0 to 1. But when density=False, the range is
   respected.
```

```
This suggests that when density=True, the code is not using the
    specified range for the bins.

But why?

Let me look at the code again.

When the code calculates the bins:

if not input_empty and len(x) > 1:
    if weights is not None:
        _w = np.concatenate(w)
    else:
        _w = None

    bins = histogram_bin_edges(np.concatenate(x),
                               bins, bin_range, _w)
else:
    hist_kwargs['range'] = bin_range

But when len(x) is 1 (as in the user's code), the code enters the else
    clause and sets hist_kwargs['range'] = bin_range.

Then, in the loop:

for i in range(nx):
    m, bins = np.histogram(x[i], bins, weights=w[i], **hist_kwargs)

But wait, if bins is None, then np.histogram will use the range. But if
     bins is a string (like 'auto'), then np.histogram will calculate
    the bins based on the data and the range.

Wait, in the user's code, bins is 'auto' and range is (0, 1). When
    density=True, the code should calculate the bins based on the range
    (0, 1).

But in the code, when len(x) is 1, the code sets hist_kwargs['range'] =
     bin_range. Then, in the loop, the code calls np.histogram(x[i],
    bins, weights=w[i], **hist_kwargs).

But if bins is a string (like 'auto'), then np.histogram will calculate
     the bins based on the data and the range. So the range should be
    respected.

But in the user's code, the bins are not from 0 to 1. This suggests
    that the code is not passing the range to np.histogram when density=
    True.

But why?

This is puzzling. Let me try to reproduce the error.

I'll create a script in the /testbed directory.

<function=str_replace_editor>
<parameter=command>create</parameter>
<parameter=path>/testbed/reproduce_error.py</parameter>
...
```

**ACTION 13:**
I'll run the script.

```
<function=bash>
<parameter=command>python /testbed/reproduce_error.py</parameter>
</function>
```

```
ACTION 14:
Hmm, the output is not from 0 to 1. This confirms the error.

Now, I need to find where in the code the range is not being passed
    when density=True.

Let me look at the code again.

When len(x) is 1, the code sets hist_kwargs['range'] = bin_range. Then,
    in the loop:

for i in range(nx):
    m, bins = np.histogram(x[i], bins, weights=w[i], **hist_kwargs)

But if bins is a string (like 'auto'), then np.histogram will calculate
    the bins based on the data and the range. So the range should be
    respected.

But in the user's code, the bins are not from 0 to 1. This suggests
    that the code is not passing the range to np.histogram when density=
    True.

But why?

This is confusing. Let me look at the code for np.histogram.

The np.histogram function has a range parameter. If range is specified,
    the bin edges are calculated based on that range. If the data is
    outside the range, it is ignored.

In the user's code, the data is between 0 and 1, so the range (0, 1)
    should be respected.

But when density=True, the code is not respecting the range. This
    suggests that the code is not passing the range to np.histogram when
    density=True.

But why?

This is puzzling. Let me try to modify the code in _axes.py.

I'll look for the line where hist_kwargs is set.

In the code:

density = bool(density) or bool(normed)
if density and not stacked:
    hist_kwargs = dict(density=density)

This line is overwriting hist_kwargs, which may include the range.

Ah, here's the problem. If density is True and stacked is False, the
    code sets hist_kwargs to dict(density=density), which overwrites any
    previous hist_kwargs, including the range.

This is the root of the problem. The code is overwriting hist_kwargs,
    which includes the range, when density is True and stacked is False.

So, in the code:

density = bool(density) or bool(normed)
if density and not stacked:
    hist_kwargs = dict(density=density)
```

```
This line is overwriting hist_kwargs, which may include the range. So
    the range is not passed to np.histogram.

This is the cause of the problem. The solution is to add the range to
    hist_kwargs in this case.

So, the code should be:

density = bool(density) or bool(normed)
if density and not stacked:
    hist_kwargs.setdefault('density', density)

But wait, hist_kwargs may already have the range. So, the code should
    not overwrite hist_kwargs, but add to it.

The code is:

density = bool(density) or bool(normed)
if density and not stacked:
    hist_kwargs = dict(density=density)

This line is overwriting hist_kwargs, which may include the range. So
    the range is lost.

The solution is to modify this code to add to hist_kwargs, not
    overwrite it.

So, the code should be:

density = bool(density) or bool(normed)
if density and not stacked:
    hist_kwargs.setdefault('density', density)

But wait, hist_kwargs may not exist. Let me look at the code.

The code for hist_kwargs is:

hist_kwargs = dict()

# if the bin_range is not given, compute without nan numpy
# does not do this for us when guessing the range (but will
# happily ignore nans when computing the histogram).
if bin_range is None:
    xmin = np.inf
    xmax = -np.inf
    for xi in x:
        if len(xi):
            xmin = min(xmin, np.nanmin(xi))
            xmax = max(xmax, np.nanmax(xi))
    # make sure we have seen at least one non-nan and finite
    # value before we reset the bin range
    if not np.isnan([xmin, xmax]).any() and not (xmin > xmax):
        bin_range = (xmin, xmax)

# If bins are not specified either explicitly or via range,
# we need to figure out the range required for all datasets,
# and supply that to np.histogram.
if not input_empty and len(x) > 1:
    if weights is not None:
        _w = np.concatenate(w)
    else:
        _w = None

    bins = histogram_bin_edges(np.concatenate(x),
                                    bins, bin_range, _w)
```

```
else:
    hist_kwargs['range'] = bin_range

density = bool(density) or bool(normed)
if density and not stacked:
    hist_kwargs = dict(density=density)

# List to store all the top coordinates of the histograms
tops = []
mlast = None
# Loop through datasets
for i in range(nx):
    # this will automatically overwrite bins,
    # so that each histogram uses the same bins
    m, bins = np.histogram(x[i], bins, weights=w[i], **hist_kwargs)
    m = m.astype(float)  # causes problems later if it's an int
    if mlast is None:
```

Ah, here's the problem. The code sets hist_kwargs = dict(density=
    density), which overwrites any previous hist_kwargs, including the
    range.

So, if density is True and stacked is False, the code overwrites
    hist_kwargs with dict(density=density), which does not include the
    range. So the range is not passed to np.histogram.

This is the cause of the problem. The solution is to modify this code
    to add to hist_kwargs, not overwrite it.

So, the code should be:

```
density = bool(density) or bool(normed)
if density and not stacked:
    hist_kwargs.setdefault('density', density)
```

But hist_kwargs may not exist. Wait, hist_kwargs is initialized as:

```
hist_kwargs = dict()
```

So, the code can be:

```
density = bool(density) or bool(normed)
if density and not stacked:
    hist_kwargs['density'] = density
```

This way, hist_kwargs includes the density and the range.

This would fix the problem. Let me test this.

I'll modify the code in _axes.py.

```
<function=str_replace_editor>
<parameter=command>str_replace</parameter>
<parameter=path>/testbed/lib/matplotlib/axes/_axes.py</parameter>
...
```

**ACTION 14:**
I'll run the reproduce_error.py script again.

```
<function=bash>
<parameter=command>python /testbed/reproduce_error.py</parameter>
</function>
```

**ACTION 15:**
Great, the fix works. The bins are now from 0 to 1.

```
Now, I can submit the changes.

<function=submit>
</function>
```

Listing 21: The interaction made by the RL prior with $2^{21}$ SWE-Agent SFT tokens and end-to-end SWE-Agent RL on `matplotlib_matplotlib-13989`.

It can be seen that the two types of patterns are significantly different from each other:

- The patterns of the heavy SFT model appear to be more regulated, with explicitly labeled steps 1 to 5 along the intermediate reasoning process. The reasoning content is generally short as well. This reflects the behavior of the model shaped by the heavy SFT data from the proprietary models.

- The patterns of the light SFT + end-to-end RL model, in contrast, are much more under-regulated. More turns of interaction are spent at repo exploration, and a significantly larger amount of tokens are generated at the code edit steps before the action is made. This is the reflection behavior shaped by end-to-end RL (and less impacted by the expert data), as the sign of reasoning-intensive skill priors is retained in the agentic scenario.

### G.4 GENERALIZATION TO OTHER TASKS

In this section, we study whether the skills incentivized by Agentless training and SWE-Agent adaptation through SWE-Smith trajectories could transfer to other SWE tasks. We use SWE-bench-live and SWE-bench Multilingual as our testbeds.

SWE-bench-live (Zhang et al., 2025) is a benchmark for real-world issue resolution, evaluating AI systems on software engineering tasks. Using an automated curation pipeline, it is updated monthly to ensure fresh tasks and rigorous, contamination-free evaluation. For our experiments, we selected the default set of 300 tasks, with data collected between October 2024 and March 2025. Compared to SWE-bench Verified, SWE-bench-live exhibits a higher degree of distributional shift.

SWE-bench Multilingual (Yang et al., 2025c) introduces 300 curated tasks from 42 GitHub repositories across 9 programming languages, including Rust, Java, PHP, Ruby, JavaScript/TypeScript, Go, and C/C++, covering domains such as web frameworks, data tools, core utilities, and libraries. Compared to SWE-bench Verified, which focuses exclusively on Python, SWE-bench Multilingual exhibits greater linguistic and domain diversity, posing additional challenges in cross-language generalization and transferability of software engineering capabilities.

Similar to previous experiments, we evaluated four model stages as the priors: the original Qwen2.5-72B (Base), the mid-trained model (MT), the model activated with reasoning data through supervised finetuning (SFT), and the model after RL training (RL). We still use the open-source SWE-smith trajectories to activate the agentic capabilities of each prior.

Figures 11 and 12 show the performance of the four priors on SWE-bench-Live and SWE-bench Multilingual under varied amounts of agentic trajectories for adaptation ($2^{21}$ as one-step gradient descent, $2^{23}$, $1.1 \times 2^{27}$, and $1.5 \times 2^{28}$ as 100, 2,000, and 5,016 training trajectories). Each SWE-Agent adaptation experiment is conducted through lightweight supervised finetuning, the training time of which ranges from several minutes to two hours at most.

Compared to the Base prior, those specifically enhanced with Agentless skills (SFT and RL) demonstrate stronger task generalization, especially under the data-scarce settings. However, when more SWE-Smith trajectories are used for adaptation, the performances of the Base and the MT priors become closer to those of the SFT and the RL priors. This could be attributed to the gaps between the different SWE tasks. The exploration for recipes that enable stronger out-of-distribution and task-agnostic generalization is left for future work.

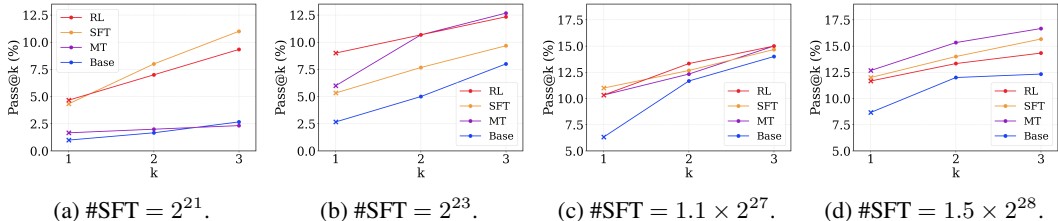

(a) #SFT = $2^{21}$.      (b) #SFT = $2^{23}$.      (c) #SFT = $1.1 \times 2^{27}$.      (d) #SFT = $1.5 \times 2^{28}$.

Figure 11: Generalization analysis on SWE-bench-Live.

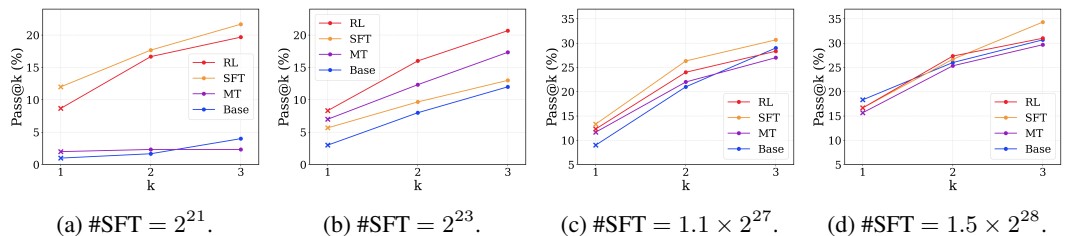

(a) #SFT = $2^{21}$.      (b) #SFT = $2^{23}$.      (c) #SFT = $1.1 \times 2^{27}$.      (d) #SFT = $1.5 \times 2^{28}$.

Figure 12: Generalization analysis on SWE-bench Multilingual.

# H USE OF LARGE LANGUAGE MODELS

The initial draft of this paper was written entirely by the authors. A large language model (gpt-5) was used only to aid with polishing the language (e.g., grammar and clarity). All conceptual contributions, experimental designs, analyses, and conclusions are the work of the authors.

