# OpenReview forum: "Kimi-Dev: Agentless Training as Skill Prior for SWE-agents"
_ICLR.cc/2026/Conference — ICLR 2026 Poster_

### Official Review · Reviewer_KmTh · 2025-10-31

**Soundness:** 3
**Presentation:** 3
**Contribution:** 3
**Rating:** 8
**Confidence:** 4

**Summary:**

This paper proposes a two-stage training approach for software engineering (SWE) agents: first training on "Agentless" workflows (fixed pipeline tasks such as bug localization and repair), then adapting to multi-turn agentic frameworks. The authors introduce Kimi-Dev, a 72B parameter model that achieves 60.4% on SWE-bench Verified through Agentless training, and 48.6% when adapted to the SWE-Agent framework with only 5k trajectories.

The central thesis is that Agentless training, framed as a single-turn task, facilitates more scalable RL training and serves as an effective prior for transitioning to multi-turn scaffoldings.

**Strengths:**

1. The paper is well-written with clear separation and explanation of each component, making the methodology easy to understand and follow.
2. The ablation studies (Figure 5) systematically isolate the contributions from mid-training, cold-start, and RL stages, providing clear illustration of each component's contribution. The sweep over different amounts of SWE-Agent SFT data (from 0 to ~1.5×2^28 tokens) is particularly comprehensive.
3. The reported performance of 60.4% on SWE-bench Verified within the Agentless scaffolding using Qwen2.5-72B as the base model is impressive.
4. The extensive appendices document data recipes, training procedures, and infrastructure details in substantial depth, significantly enhancing the work's reproducibility.

**Weaknesses:**

1. The anomalous performance dip of the "SFT" prior at 2^24 tokens (Figure 5) raises questions about the stability and reliability of the proposed priors. Including confidence intervals would help clarify whether the observed differences between priors are statistically significant or within the margin of experimental noise.
2. The 5k SFT trajectories represent a relatively small subset of potential teacher data. Given that Claude 3.7 Sonnet is a substantially stronger model, there may be a scenario where scaling the number of trajectories could narrow the gap between the Base and Kimi-Dev priors. If this is the case, the proposed approach would offer a more efficient rather than more powerful method for training multi-turn SWE agents, which represents a weaker claim that it could be otherwise.
3. While the Agentless checkpoint demonstrates strong performance as a prior for multi-turn adaptation, it remains unclear whether directly applying large-scale RL to the SWE-Agent framework would yield comparably strong or weaker results. The literature includes examples of applying GRPO (DAPO) to sparse reward settings for LLMs in SWE tasks, such as DeepSWE [1]. A direct comparison or discussion of this alternative training paradigm would strengthen the paper's claims about the necessity and superiority of the two-stage approach.

[1] Michael Luo et al. DeepSWE: Training a State-of-the-Art Coding Agent from Scratch by Scaling RL

**Questions:**

1. What is the precise configuration of SFT used for Figure 5? While the number of trajectories is clearly specified, what is the number of training epochs? Were hyperparameters tuned independently for each experimental run, or was a single configuration used across all conditions?
2. What are your thoughts on the TestWriter false positive examples documented in the appendix, and how might this limitation be addressed in future work?

---

> ### Author Response · Authors · 2025-12-03
> **Rebuttal by Authors**
>
> Thank you for your insightful suggestions!
>
> ***Weakness 1 & Question 1. The anomalous performance dip of the SFT prior at 2^24 tokens (Figure 5) raises questions about the stability and reliability of the proposed priors. Besides, what is the precise configuration (e.g. the training epochs) of SFT used for Figure 5?***
>
> We first provide the precise configuration of SFT used for Figure 5: we use full-parameter SFT in all the data settings for 3 epochs with 128 GPUs (16 nodes). We use a fixed constant learning rate of 1e-6 and a maximum length of 64k for all experiments. We set tensor/pipeline/context parallel size to be 2/8/4, respectively. In all the experiments except for the $2^{21}$ and the $2^{23}$ tokens settings, we set the global batch size to be 64. For the $2^{21}$ and the $2^{23}$ tokens settings (from the subset of 100 SWE-Smith trajectories), the amount of the SFT sequences after 64k packing is less than 64, and we set the global batch size as 32; 3 epochs over this SFT data could just be traversed within 4 steps of gradient descent, resulting in $2^{23}$ tokens.
>
> The reason of the performance dip of the SFT prior at $2^{24}$ tokens is therefore inferred to be the doubled global batch size from 32 to 64 at the $2^{24}$ tokens setting, as an enlarged batch size may exacerbate overfitting under such efficient transfer conditions. Nevertheless, we believe this does not affect the overall trends for different priors demonstrated in Figure 5, as starting from the $2^{24}$ tokens setting, the global batch size together with all of other hyperparameter settings is kept as the same, and no performance dip is further observed.
>
>
> ***Weakness 2. Will scaling up the amount of trajectories distilled from teacher models narrow the gap between Base and Kimi-Dev priors, and how this affects the conclusion of the paper.***
>
> In this work, we restrict our access of expert supervision to open-sourced LLMs and publicly available datasets only. For SWE-Agent data, we rely solely on the 5k trajectories released by SWE-Smith (mostly from Claude 3.7 Sonnet).
>
> Therefore, the major contribution of the paper revolves around the settings with limited access to expert data: Under such limited setup, Agentless skill priors have demonstrate the benefits for efficient transfer to the SWE-Agent framework. Such conclusion could also be generalized to other domains / tasks where long-horizon, multi-turn expert data are expensive or difficult to collect, and we leave further exploration as future work.
>
> We also leave as future work with the setting where access is provided to more data from teacher models or human annotation, and the trade-off between skill prior and in-domain data enhancement could be studied.
>
>
> ***Weakness 3. Directly applying large-scale RL to the SWE-Agent framework would strengthen the paper's claims about the necessity and superiority of the two-stage approach.***
>
> In Page 10 of the updated paper, we've added end-to-end SWE-Agent RL experiments to further demonstrate the effectiveness of the two-stage approach. We select the setting with $2^{21}$ SWE-Agent SFT tokens (the second column in Figure 5) where a single step of gradient decent takes place on top of each prior, so that the effect from the proprietary model data is maximally alleviated.
>
> According to the results in Page 10 and Figure 7, the end-to-end RL failed for MT prior, as the amount of the initial problem set that suffices pass@8=1 for MT prior's RL is as small as 260. While for both the SFT and the RL priors, the stronger priors lead to 2,062 problems with initial pass@8=1 for RL training, and 300 steps of RL over these lead to around +15% absolute jump in pass rates. The ultimate performance of the RL prior is slightly higher than that of the SFT prior. In Appendix E.3, we also show that the patterns learned through end-to-end SWE-Agent RL differ significantly from those learned by expert data distillation. The end-to-end RL experiments demonstrate the **necessity of the priors in unlocking more problems for RL**, and the **superiority of the priors in their potential that could be incentivized further with RL**.
>
> ***Question 2. What is the potential remedy for the TestWriter's false positiveness in the future work?***
>
> As outlined in Appendix F.2, the TestWriter's false positive examples are usually attributed to their insufficient coverage, which fail to account for the test cases other than the generated one while necessary for the BugFixer to pass. A potential idea is to use the existing test coverage check tools (e.g., `pytest-cov` in Python) for more fine-grained reward design. We will experiment with this in future work.

---

### Official Review · Reviewer_kUZK · 2025-10-31

**Soundness:** 3
**Presentation:** 3
**Contribution:** 2
**Rating:** 6
**Confidence:** 4

**Summary:**

The paper introduces Kimi-Dev, an open-source LLM for software engineering that unifies agentless and agentic approaches. It shows that agentless training through modular tasks can build transferable skill priors that make subsequent agentic finetuning more efficient. Kimi-Dev achieves SOTA on SWE-bench Verified and enables competitive SWE-agent results with minimal additional training.

**Strengths:**

The main discovery of agent less training can induce skill that make further agent training more effective is interesting: it opens up a new paradigm how researchers may train their agent. The open source of the recipe further add to the contribution.
The proposed multi-stage training recipe is well-explained and reflect the researcher’s understanding of the nuances in scaling SWE models.
The empirical results of the trained model is strong, achieving SOTA among open-sourced models for workflow-based approaches on SWE-bench, and competitive pass@1 performance under agentic framework. This shows the method’s effectiveness.
The analysis of the paper is very insightful, including the varying of token budget, RL progression, etc. All of these settings reflects a solid experimental approach.

**Weaknesses:**

Some technical details are deferred to the appendix. The main context only gives a final evaluation formula while the concrete implementations of reward, prompt sampling, etc. are not rigorously and clearly laid out in the main text.
There are some observations about false positive mentioned, but the paper can further benefit from providing quantitative and qualitative analysis of the error remaining given the current bug fixer and test writer, especially points that can reflect the challenges in agentic skill prior transfer.
Much of the justification for skill transfer from agent less to agentic framework is empirical. As the core discovery of the paper, more principled proof or study about how skill prior linked to generalization and how these skills are manifested will further strengthen the claim.

**Questions:**

N/A

---

> ### Author Response · Authors · 2025-12-03
> **Rebuttal by Authors**
>
> Thank you for your valuable suggestions!
>
>
> ***Weakness 1. Some technical details (reward, prompt sampling, etc.) are deferred to the appendix.***
>
> We've incorporated more technical details in the main text of the updated paper for a clearer description regarding the reward  design and the prompt sampling strategy. We will also polish the paper by further refactoring the important information from appendix to the main body.
>
> ***Weakness 2. Provide quantitative and qualitative analysis of the error remaining given the current bug fixer and test writer, especially points that can reflect the challenges in agentic skill prior transfer.***
>
> We analyze the set of the passed / failed instances of several SWE-Agent models by taking the intersection with the sets of passed / failed instances of the BugFixer and correctly / incorrectly reproduced instances of the TestWriter, and study the count statistics for quantitative analysis.
>
> For BugFixer and TestWriter, we use the RL prior under the Agentless framework (48.0% pass@1 with temp=0 for BugFixer, and 37.6% correct reproducing rate with temp=0 for TestWriter).
>
> For SWE-Agent models, we select
> (i) the RL prior with $1.5\times2^{28}$ SWE-Agent SFT tokens (48.6% pass@1 with temp=0),
> (ii) the RL prior with $2^{21}$ SWE-Agent SFT tokens (23.0% pass@1 with temp=0),
> (iii) the RL prior with $2^{21}$ SWE-Agent SFT tokens + end-to-end SWE-Agent RL (42.0% pass@1 with temp=0; we've added the end-to-end SWE-Agent RL experiments in Page 10 of the updated paper).
>
> The intersection count analysis is shown as follows, where for each passed / failed instance from each SWE-Agent model, its pass/fail status in BugFixer AND TestWriter is counted:
>
> |                                                        | TOTAL | BF√TW√ | BF√TW× | BF×TW√ | BF×TW× |
> |--------------------------------------------------------|-------|--------|--------|--------|--------|
> | RL prior, $1.5\times2^{28}$ SWE-Agent SFT tokens, √                  | 243   | 100    | 92     | 19     | 32     |
> | RL prior, $1.5\times2^{28}$ SWE-Agent SFT tokens, ×                  | 257   | 16     | 32     | 53     | 156    |
> |                                                        |       |        |        |        |        |
> | RL prior, $2^{21}$ SWE-Agent SFT tokens, √                           | 115   | 52     | 44     | 11     | 8      |
> | RL prior, $2^{21}$ SWE-Agent tokens, ×                           | 385   | 64     | 80     | 61     | 180    |
> |                                                        |       |        |        |        |        |
> | RL prior, $2^{21}$ SWE-Agent SFT tokens + end-to-end SWE-Agent RL, √ | 210   | 94     | 82     | 16     | 18     |
> | RL prior, $2^{21}$ SWE-Agent SFT tokens + end-to-end SWE-Agent RL, × | 290   | 22     | 42     | 56     | 170    |
>
> According to the statistics, the performance of BugFixer shows a higher correlation with the SWE-Agent performance in all the three settings: Among the passed instances of SWE-Agent, the counts with BugFixer passing and TestWriter correct are similar to that with BugFixer passing and TestWriter wrong, and are significantly more than the counts with BugFixer wrong and TestWriter correct. Qualitatively, we have added case studies by comparing the traces of the RL prior under Agentless setting and the interaction trajectories of the corresponding SWE-Agent adapted models in Appendices E.2 and E.3.
>
> The straightforward conclusion is to enhance the prior in BugFixer ability, as intuitively, a perfect BugFixer could lead to straight-shot success of code-edit even under the SWE-Agent framework. Nevertheless, as shown in Figure 6 (right), the reflection skills play an important role in the SWE-Agent, which could be related with TestWriter ability, as the error introduced in the SWE-Agent's bugfix steps is to be self-realized by writing tests. We will provide a more in-depth analysis and develop stronger priors in the future.
>
>
> ***Weakness 3. More principled proof or study about how skill prior linked to generalization and how these skills are manifested will make up for the current empirical justification.***
>
> While the motivation of this work is to unify the dichotomy between the workflow-based Agentless and the end-to-end SWE-Agent frameworks by recasting Agentless training as constructing skill priors, the idea of solving long-horizon tasks with loosely-coupled subtask structure by skill reuse with shorter horizon can be found in classical hierarchical and meta RL literature [1]. We leave further exploration along this direction as future work.
>
>
> [1] Relay Policy Learning: Solving Long-Horizon Tasks via Imitation and Reinforcement Learning. CoRL 2019.

---

### Official Review · Reviewer_xDcT · 2025-11-01

**Soundness:** 3
**Presentation:** 3
**Contribution:** 4
**Rating:** 8
**Confidence:** 5

**Summary:**

The paper argues that “agentless” workflow training can serve as skill priors (localization, code edit, self-reflection) for end-to-end SWE agents. It introduces Kimi-Dev, a 72B open-weight code LLM trained via a curated recipe (mid-training on PR/issue data, a reasoning-heavy cold start, RL on code edits, and test-time self-play). Under a workflow (agentless) setup, Kimi-Dev reaches 60.4% on SWE-bench Verified; with 5k public SWE-Agent trajectories for light SFT, the model attains 48.6% pass@1 in a multi-turn agent framework, comparable to Claude 3.5 Sonnet (241022). These results aim to bridge workflow and agentic paradigms by treating agentless training as a foundation for agent adaptation.

**Strengths:**

- This paper exercises a principled recipe and delivers strong results under agentic settings. The staged recipe (mid-training on ~150B tokens; 50B natural diffs, 20B PR commit packs, ~20B synthetic, upsampled ×4; reasoning-heavy cold start; RL with outcome-only rewards; test-time self-play) is coherent and empirically validated. The paper achieves 60.4% on SWE-bench Verified under a standardized 40 patch/40 test setting, with clear scoring for self-play selection.
- Agent transfer with small data. With only ~5k public SWE-Agent trajectories, the adapted model attains 48.6% pass@1
- The authors performed detailed experiments demonstrating the skill transfer and generalization effect by controlling for different prior models. These experiments provide valuable insights into how the community can improve future agent models.

**Weaknesses:**

- It is unclear to me that RL is an indispensable part of the recipe: In Figure 5, the performance difference between the RL and the SFT curves is relatively small. Does this demonstrate that the RL part is not really the essential part of the recipe—SFT on an agentless task is already sufficient to provide such a prior to improve the upper bound on performance given sufficient multi-turn agent SFT data?
- Unclear if Agentless Skill prior will still be useful given sufficient multi-turn SFT data: Also in Figure 5, when given sufficient agent SFT data, the performance of MT gets very close to SFT/RL. Does that mean the value of agentless skill prior drops as you have access to more in-domain agent SFT data?
- Unclear if performance improvements come from distilling from a stronger model: If I understand correctly, the SFT data for agentless skill was distilled from Deepseek R1 (section 3.2) - could the effectiveness of "agentless prior" come from distillation from the Deepseek R1 model?

**Questions:**

- For weakness 1: it will nice to either (1) show the benefit of RL by demonstrating performance improvements on other datasets/metrics/settings (eg higher pass@k for large k); OR (2) discuss the limitation of agentless+RL in providing better prior
- For weakness 2: It will be nice to either (1) disprove it or (2) discuss the limitation
- For weakness 3, my question is: Given the same computation budget, what if we perform distillation on the same set of training tasks (eg, swe-smith, swe-gym) -- Is it possible that the performance gain we get there will be larger than the one we get with agentless as prior? If so, why is it a good idea to use an agentless prior rather than directly generating data from environments that better match the domain?

---

> ### Author Response · Authors · 2025-12-03
> **Rebuttal by Authors**
>
> Thank you for your supportive review!
>
> ***Weakness & Question 1. Unclear about the role of RL in the Agentless training recipe as providing prior - since the performance gap between the SFT prior and the RL prior is close.***
>
> Following the option (1) listed in Question 1, we've added end-to-end SWE-Agent RL experiments on top of different priors with the setting of $2^{21}$ SWE-Agent SFT tokens (extremely lightweight SFT over the priors). According to the performance in Figure 7 of the revised paper, the SWE-Agent performances (average over 5 runs, pass@3, pass@5) by the RL prior slightly surpass those from the SFT prior. This suggests that the skill priors enhanced by Agentless RL could transfer to the SWE-Agent performance as well. However, the transferred superiority of RL over SFT under SWE-Agent is narrower than that under Agentless, suggesting alternative ways to allocate compute (e.g. scaling SWE-Agent RL with Agentless SFT prior) could be more optimal from the utility perspective (as option (2) suggests).
>
> ***Weakness & Question 2. Unclear if the Agentless skill prior will still be useful given sufficient multi-turn SFT data.***
>
> In this work, we restrict our access of expert supervision to open-sourced LLMs and publicly available datasets only. For SWE-Agent data, we have access to the 5k trajectories released by SWE-Smith (mostly from Claude 3.7 Sonnet). Under this limited setup, Agentless skill priors lay a solid foundation for efficient transfer to the SWE-Agent framework.
>
> Using more expert annotation or frontier model distillation data for the multi-turn SWE-Agent setting could be viewed as directly enhancing the SWE-Agent ability itself, and we leave the trade-off study between skill prior and in-domain data enhancement in future work.
>
>
> ***Weakness & Question 3. Unclear if performance improvements come from distilling from a stronger model, and if the Agentless skills come from DeepSeek-R1; If so, will directly distilling more SWE-Agent trajectories (instead of distilling Agentless skills) yield better performance, and why do we need to construct such skill prior?***
>
> Regarding the impact from DeepSeek-R1, we refer to the response to ***Question 2*** from **Reviewer jxnV** that the major performance improvement is brought by the Agentless RL and test-time self-play stages (36.6% --> 48.0% --> 60.4%).
>
> Regarding the potential performance gain by distilling from proprietary models that excel on SWE-bench-style tasks, we refer to our response to ***Weakness & Question 2*** for future study on the comparison between prior construction and direct distillation.
>
> As for the purpose of constructing skill priors, we note that the conclusion drawn in this work on the SWE-bench domain could be generalized to other domains / tasks where long-horizon, multi-turn data with high quality are expensive or difficult to collect: In such scenario, strong skill priors could ease the burden of in-domain data requirements, as the skills need not to be learned but only to be steered and assembled. We plan to explore further in this direction in future work.

---

### Official Review · Reviewer_jxnV · 2025-11-01

**Soundness:** 3
**Presentation:** 3
**Contribution:** 2
**Rating:** 6
**Confidence:** 4

**Summary:**

This work presents a training recipe that leverages workflow-based pipelines as opposed to more open-ended agentic rollouts in order to train an open-source code LLM for SWE tasks. The so-called Agentless training approach provides more stability and the opportunity to introduce intermediate skills, such as locating files, writing tests and fixing bugs, which may subsequently be used and composed together when training a SWE-agent model. Concretely, the authors describe: a mid-training procedure where data from GitHub issues and PR commits form examples of how developers reason about issues and fix bugs; a cold-start procedure where SWE-related reasoning traces from R1 are used to induce long CoT abilities; an RL stage where a policy gradient objective is used to train code editing capabilities with outcome-based rewards; and finally a test-time self-play stage where the model learns to play the roles of a BugFixer and TestWriter. Following this training procedure, the authors evaluate the resulting Kimi-Dev model on SWE-Bench Verified, achieving a strong performance of 60.4% for an open-weights 72B model derived from Qwen 2.5 (Base). The authors then investigate whether the structured Agentless training procedure leads to model checkpoints which offer useful "priors" for more open-ended SWE-agent fine-tuning. Indeed, various stages of the Agentless training procedure, in particular the mid-training, lead to a big uplift in SWE-Bench Verified performance after various amounts of SFT on SWE-Agent traces collected from Claude 3.5 Sonnet. The authors also demonstrate some evidence towards the RL prior (after the full Agentless training procedure) supporting a greater number of turns for SWE-Agent style rollouts after SFT.

**Strengths:**

While advanced coding agent model training has been underway in model labs for certain years, there are relatively few detailed accounts of how to take a pre-trained model and effectively imbue it with the skills needed to solve software engineering problems. From this perspective, the work offers a valuable contribution, in particular the details of the mid-training procedure which appears to be the stage of Agentless training that yields the largest SWE-Bench Verified performance improvement after SFT on SWE-Agent traces; perhaps due to training the model to reproduce the motifs, reasoning patterns and skills required to conduct software engineering work. The work is of a reasonably large scale, and achieves compelling results on SWE-Bench Verified for a model of its size, with 60.4% resolution rate on SWE-Bench Verified with the Agentless-like framework.

**Weaknesses:**

While the Agentless performance for SWE-Bench Verified is reasonably strong at 60%, the pass@1 rate on the SWE-Agent results are somewhat low at 48.6%, particularly given the model is roughly double the size of other models obtaining comparable scores. It would be good to investigate why this is, and crucially whether this is as a result of the Agentless training procedure that preceded SFT for SWE-Agent.

Moreover, the authors train on the outputs of other models at two key points in the paper. This is disappointing since it effectively hides how the corresponding skills can be trained from scratch in an open-source model, instead deferring this to the black box of a preexisting model trained by some other lab. The first instance of this is during mid-training - which appears to be one of the steps in the training procedure which yields the greatest performance improvements - where traces from the DeepSeek's R1 model are used to introduce skills such as reasoning, problem analysis, method sketching, self-refinement and exploration of alternative solutions. This is an important procedure, and it would have been good to see discussion of how these skills could be bootstrapped without training on the outputs of a model which has already undergone this training. The second instance are the SWE-Agent traces used for SFT from Claude 3.5 Sonnet. While I understand that the primary focus of this paper is the Agentless training procedure itself, I do believe that any conclusions pertaining to the usefulness of this intermediate training procedure as a "prior" for training SWE-Agents must be made with a more legible training procedure for the SWE-Agent stage. In other words, had the jump from Agentless to SWE-Agent been achieved by a transparent RL procedure, then conclusions about the usefulness of the Agentless skill prior may be more impactful. Instead, we appear to have learned that these "priors" are useful when doing imitation learning from black-box model traces.

Finally, on the point about generalization, the studies all revolve around SWE-Bench style tasks (including the live and multilingual variants in Appendix E). These represent only a small subset of software engineering work, and do not address related tasks such as novel algorithm design, performance optimization, task management and so forth, which I believe would be a better quantifier of generalization than merely more up-to-date or polyglot variants of SWE-Bench's issue resolution task. Indeed, this is an area where the paper's Agentless training thesis may hold promise; where systematically created workflow-based mid-training procedures can imbue a model with a diverse range of useful skills which can be learned to be composed together in the RL stage.

Nits:
- L427: do you mean to say "The RL prior outperforms all the other models..."?

**Questions:**

- what is behind the roughly linear increase in token length in Figure 3 as RL training proceeds? You analyze some of these traces in Appendix F, observing that there are some repetitive self-reflections, although also noting that increased response length correlated with improved answer accuracy. How do you quantify this at scale? Is the length increase due to your choice of RL objective? Recent work (https://arxiv.org/pdf/2508.08221) has commented on this sequence length effect, albeit with a GRPO-style objective.
- How much do you attribute the performance improvements measured in both the Agentless framework evaluations and subsequent SWE-Agent evaluations to the dataset of DeepSeek R1 based reasoning trajectories? Looking at Figure 5, after $2^{23}$ SWE-Agent SFT tokens there is a large jump between the base model and MT checkpoint. What explains this?

---

> ### Author Response · Authors · 2025-12-03
> **Rebuttal by Authors (1/3)**
>
> Thank you for your valuable feedback!
>
> ***Weakness 1. Why does the model suffer a performance drop under SWE-Agent （\~48%） compared with that under Agentless （\~60%）?***
>
> As introduced in Section 2.1 and Section 4, compared with Agentless, the SWE-Agent framework poses special challenges as it requires the agent to follow a more complex SWE-Agent system prompt, complete the task in an end-to-end fashion, with multi-turn, long-horizon execution from repo exploration to self-verfication with test edgecases. This makes the SWE-Agent setting inherently more difficult to optimize with.
>
> Despite the higher level of difficulty, it is noted that the **48.6%** pass rate under SWE-Agent is achieved with **a single attempt**, while the 60.4% pass rate under Agentless is obtained with test-time self-play in a 40-patch, 40-test setting. Figure 4 shows that in a **single-attempt Agentless** setting, the model has a pass rate of **48.0%**. This reveals that under the single attempt setting, the model's adaptation to SWE-Agent results in similar pass rates on SWE-bench Verified. For a fair comparison with the 60.4% pass rate under Agentless test-time self-play, voting mechanisms / verifier models proposed by [1][2] should be leveraged, which have demonstrated significant improvement over the single attempt performance (in our case, 48.6%); We leave this as future work.
>
> [1] R2e-gym: Procedural environments and hybrid verifiers for scaling open-weights swe agents.
>
> [2] Training software engineering agents and verifiers with swe-gym.
>
> ***Weakness 2. As similar SWE-Agent performance is achieved by other models with nearly half the parameter size, can this inferiority of Kimi-Dev be the consequence of Agentless training?***
>
> The Agentless training process benefits the SWE-Agent adaptation instead of bringing inferiority. The conclusion is made from Figure 5, where the SFT prior and the RL prior imbued from Agentless training significantly outperform the Base prior with the same amount of SWE-Agent tokens for SFT.
>
> As for comparison with other 32B models in Table 2, it is noted that the starting points of the models they used to train **differ** from the one we use. For example, SWE-agent-LM uses QWen2.5-Coder-32B-Instruct as the starting point, while DeepSWE uses QWen3-32B-Instruct. In our work, the starting point we use is QWen2.5-72B-**Base**, and we have developed the entire mid-training and post-training recipes for SWE-bench, demonstrate performances on different frameworks, and contribute to skill generalization. According to the pass rates in Table 2, our models outperform all the 32B counterparts by an at least 6% absolute margin.
>
> While a fair comparison could not be made directly to explain for the superiority of our model in Table 2, conclusion could be made by comparing the performance of the 5k SFT trajectories over different priors (shown in **Figure 5, the rightmost column**) with that of SWE-agent-LM (where the same 5k trajectories were used from SFT over QWen2.5-Coder-32B-Instruct): The performance from SWE-agent-LM stays comparable to the Base prior, while outperformed by the MT, SFT, and RL priors. This again proves the efficacy of the Agentless training stage for skill prior enhancement.
>
>
> ***Weakness 3. Explain the role of the traces distilled from DeepSeek-R1 in the mid-training phase.***
>
> As the reviewer expresses concern about "training on the output of other models", we first state the principle regarding this that we sticked to: In this work, we restrict our access of expert supervision to **open-sourced LLMs** and **publicly available datasets** only. Akin to the use of human-annotated data in the development of frontier models, we perform lightweight supervised finetuning for cold-start use, and put the emphasis of our contribution on other stages including mid-training, RL, skill transfer and generalization.
>
> For DeepSeek-R1 data, we construct a dataset of 4k BugFixer and 4k TestWriter traces for the cold-start of the Agentless code-edit phase. For mid-training, as outlined in Section 3.2 and detailed in Appendix A, nearly 80% of the tokens in the data recipe are natural data consisting of natural diff patches and PR commit packs processed from GitHub crawl. It should be also noted that the greatest performance improvement does not come from such distillation process, but is brought by Agentless RL (36.6% --> 48.0% BugFixer pass@1) and test-time self-play (48.0% BugFixer pass@1 --> 60.4% final@1) stages. R1 is not involved in these stages.

---

> > ### Author Response · Authors · 2025-12-03
> > **Rebuttal by Authors (2/3)**
> >
> > ***Weakness 4. SWE-Agent traces used for SFT come from Claude 3.5 Sonnet, which affects the conclusion related to the Agentless skill prior. The usefulness of the prior may be more impactful if the jump from Agentless to SWE-Agent is achieved by a transparent RL procedure.***
> >
> > In Page 10 of the revised paper, we've added end-to-end SWE-Agent RL experiments to further demonstrate the effectiveness of different Agentless priors. To maximally alleviate the effect from the proprietary model data within the SWE-Smith trajectories, we leverage the setting with $2^{21}$ SWE-Agent SFT tokens (the second column in Figure 5) where **a single step of gradient decent** takes place on top of each prior. We use this minimal cold-start setup to nurture the performance jumps of different priors brought by RL as transparent as possible.
> >
> > According to the results in Page 10 and Figure 7, the end-to-end RL failed for the MT prior, as the amount of the initial problem set that suffices pass@8=1 for MT prior's RL is as small as 260. While for both the SFT and the RL priors, the stronger priors lead to 2,062 problems with initial pass@8=1 for RL training, and 300 steps of RL over these lead to around +15% absolute jump in pass rates. The ultimate performance of the RL prior is slightly higher than that of the SFT prior. In Appendix E.3, we also show that the patterns learned through end-to-end SWE-Agent RL differ significantly from those learned by expert data distillation. The end-to-end RL experiments demonstrate the usefulness of the priors in both unlocking more problems for RL and their potential that could be incentived further with RL.
> >
> >
> > ***Weakness 4. Generalization studies revolve around SWE-Bench style tasks only; though Agentless training may hold promise for other task types.***
> >
> > As we dedicate the training recipe in this work for SWE-bench, we selected SWE-bench Multilingual and SWE-bench Live to study the generalization abilities of the trained model in terms of programming languages and issue dates.
> >
> > Nevertheless, we appreciate that the reviewer acknowledges the promise of Agentless training to generalize over more task types beyond SWE-bench style tasks. We also agree that the broader sense of task generalization could be achieved by generalizing the skill-based mid-training and incentivizing composition during RL phase), as similar ideas are suggested in classical RL literature [1]. We leave the exploration of this direction as future work.
> >
> > [1] Relay Policy Learning: Solving Long-Horizon Tasks via Imitation and Reinforcement Learning. CoRL 2019.
> >
> > ***Nits on L427: do you mean to say "The RL prior outperforms all the other models..."?***
> >
> > Yes, thanks for pointing out. We've updated the paper in the revision.
> >
> > ***Question 1: Quantify the phenomenon of roughly linear increase in token length during RL training at scale.***
> >
> > The increase in token length is a natural consequence of reinforcement learning on top of LLMs, which has been observed in different RL objective design including the REINFORCE-style loss ([2], Figure 5), GRPO loss ([3], Figure 3), DAPO loss ([4], Figure 7-(a)), CISPO loss ([5], Figure 4), etc., where performances on different tasks are improved by scaling test-time compute through generating longer Chain-of-Thoughts. In our work on SWE-bench, we've observed similar phenomenon in both 72B and 14B experiments (Figures 3 and 9, respectively). More in-depth analysis on the length increase effect from RL objectives is out of the scope of the paper and is left for future work.
> >
> > [2] Kimi k1.5: Scaling Reinforcement Learning with LLMs.
> >
> > [3] DeepSeek-R1: Incentivizing Reasoning Capability in LLMs via Reinforcement Learning.
> >
> > [4] DAPO: An Open-Source LLM Reinforcement Learning System at Scale.
> >
> > [5] MiniMax-M1: Scaling Test-Time Compute Efficiently with Lightning Attention.
> >
> > ***Question 2: Analyze the impact of DeepSeek-R1 data on the Agentless and subsequent SWE-Agent performance.***
> >
> > Similar to the response to ***Weakness 3***, we use a small amount of data from DeepSeek R1 for cold-start use only. The major performance gains are brought by Agentless RL (36.6% --> 48.0% BugFixer pass@1) and test-time self-play (48.0% BugFixer pass@1 --> 60.4% final@1), where R1 data are not involved.
> >
> > In the subsequent SWE-Agent experiments, according to the performance sweeping in Figure 5 and analysis in Figures 6, Agentless RL always provides the most significant transferred performance and skill demonstration. We therefore acknowledge that the data from DeepSeek-R1 have played a vital role for cold-start purpose, but require other recipes including mid-training, RL, test-time self-play, and skill transfer to form a SWE-Agent model that achieves the 48.6% performance comparable with Claude 3.5 Sonnet (20241022).

---

> > > ### Author Response · Authors · 2025-12-03
> > > **Rebuttal by Authors (3/3)**
> > >
> > > ***Question 3: Explain the performance jump between the Base and the MT prior at $2^{23}$ tokens in Figure 5.***
> > >
> > > The performance jump of the MT prior with $2^{23}$ SWE-Agent SFT tokens is because this amount of SWE-Agent cold-start suffices to incentivize the skills embedded in the MT prior and differentiate it from the Base prior (while $2^{21}$ tokens are not enough).
> > >
> > > More specifically, for the MT prior experiment, when comparing trajectories after training with $2^{21}$ and $2^{23}$ SWE-AGENT SFT tokens, we observed a significant improvement in the correct use of the str_replace tool in the latter: In the MT prior with $2^{21}$ SWE-Agent SFT tokens, nearly 40% of the incorrect str_replace tool calls are due to failures such as producing incorrect values for the old_str and new_str arguments. Another roughly 40% of the errors come from cases where old_str and new_str are the same without effective change made. In contrast, after finetuned with $2^{23}$ SWE-Agent SFT tokens on top of the MT prior, these two types of str_replace errors are significantly reduced.

---

### Meta-Review · Area_Chair_W2Ji · 2026-01-07

**Summary:**

The reviewers broadly agree that the paper presents a well-designed and carefully executed training recipe that bridges agentless workflows and agentic SWE frameworks, with strong empirical performance on SWE-bench Verified. In particular, reviewers highlight the clarity of the staged training pipeline and the extensive ablations isolating the effects of mid-training, cold-start, and RL components (**[xDcT, KmTh]**).

While several reviewers raise questions about the necessity of individual components (e.g., Agentless RL vs. SFT) and the scope of the paper’s generalization claims, these concerns are framed as limitations or avenues for future work rather than fundamental flaws (**[jxnV, xDcT, KmTh, kUZK]**).

The rebuttal addresses many of these points by adding end-to-end SWE-Agent RL experiments, clarifies attribution of gains across stages, and provides additional analysis on stability and error modes (**[jxnV, xDcT, kUZK, KmTh]**).

**Reviewer Concerns:**

**1. Role and necessity of Agentless RL relative to SFT**
Some reviewers question whether the RL stage is strictly necessary, given that the performance gap between SFT and RL priors narrows under sufficient SWE-Agent SFT (**[xDcT, KmTh]**). The rebuttal partially addresses this by adding end-to-end SWE-Agent RL results showing modest but consistent gains from stronger priors (**[xDcT, KmTh]**).

**2. Attribution of gains vs. distillation from stronger models**
Multiple reviewers express concern that distillation from DeepSeek-R1 and Claude Sonnet complicates attribution of improvements to the Agentless paradigm itself (**[jxnV, xDcT]**). The authors clarify that such data is used mainly for cold-start and that the largest gains arise from Agentless RL and test-time self-play, which reviewers acknowledge as a reasonable clarification (**[jxnV, xDcT]**).

**3. Scope of generalization beyond SWE-bench-style tasks**
Reviewers note that evaluations are limited to SWE-bench and its variants, and generalization to broader software engineering is not clear (**[jxnV, xDcT]**). This limitation is explicitly acknowledged by the authors and framed as future work (**[jxnV, xDcT]**).

**4. Stability, statistical significance, and analysis depth**
Concerns are raised regading performance fluctuations (e.g., SFT prior dips) and the lack of confidence intervals ([KmTh]). The rebuttal explains the observed instability via batch size changes and adds quantitative and qualitative error analyses, which reviewers consider clarifying though not exhaustive (**[KmTh, kUZK]**).

**Reviewer Scores:**

Reviewer sentiment is overall positive. Two reviewers recommend acceptance with high confidence, citing strong empirical results, careful ablations, and the value of the proposed training recipe ([xDcT, KmTh]). The remaining reviewers give marginally positive scores and indicate they would not object to acceptance, framing their critiques as requests for clarification or broader validation rather than rejection-level concerns ([jxnV, kUZK]). No reviewer expresses strong opposition to acceptance.

---

### Decision · Program_Chairs · 2026-01-26

Accept (Poster)